# Identifying the key steps determining the selectivity of toluene methylation with methanol over HZSM-5

Qingteng Chen[1], Jian Liu [1✉] & Bo Yang [1✉]

Methylation of toluene with methanol to produce p-xylene has been investigated for decades, but the origin of selectivity is still under debate. Here we report computational studies based on ab initio molecular dynamics simulations and free energy sampling methods to identify the key steps determining the selectivity. The steps of toluene methylation to protonated-xylene, deprotonation of protonated-xylenes, and diffusion of xylene in HZSM-5 channels are compared. We find the pathways of formation for protonated p-/m-xylenes have similar free energy barriers. Meanwhile, the methylation is found rate-determining, thus the probability to generate p-/m-xylenes at the active site are similar. We then find that the diffusion for m-xylene along the zigzag channel is more difficult than its isomerization to p-xylene, which in turn further promotes the selectivity of p-xylene formation. These insights obtained at the molecular level are crucial for further development of high-performance zeolite catalysts for toluene methylation.

[1] School of Physical Science and Technology, ShanghaiTech University, Shanghai, China. ✉email: liujian@shanghaitech.edu.cn; yangbo1@shanghaitech.edu.cn

Xylene is an important raw material in the industry of synthetic fibers[1], and in particular, para-xylene (p-xylene, PX) is the most valuable product with rapidly growing demand. Currently, in the petroleum industry, xylene is mainly obtained from the processes of naphtha reforming or toluene methylation;[2,3] however, in these processes, mixed xylene containing para- (p-), meta- (m-, MX), and ortho- (o-, OX) isomers can be obtained. Therefore, enhancing the selectivity to the para-isomer in the processes is crucial.

It was reported that p-xylene can be selectively produced through alkylation or disproportionation of toluene with modified ZSM-5 zeolites as catalysts[1,4,5]. In some modified ZSM-5 zeolites, the selectivity to p-xylene can reach more than 90%[6,7], which exceeds the thermodynamic equilibrium proportion of p-xylene in xylenes, i.e., around 24%[6,8]; however, the origin of such high selectivity obtained is still under debate. Several studies suggested that the alkylation of toluene is expected to produce p-xylene as the primary product, mainly due to the stereospecificity of HZSM-5[5,9]. In contrast, alternative views[1] believed that the high p-xylene selectivity originates from the fact that the pore structure of HZSM-5 facilitates the diffusion of p-xylene[6].

When taking toluene and methanol as reactants for the production of xylene, it was reported that xylene is formed at Brønsted acid sites within HZSM-5[10–12]. There are two widely accepted mechanisms of toluene methylation: direct and stepwise mechanisms[13]. The direct mechanism suggests that protonated methanol reacts directly with toluene to produce protonated xylene, while in the stepwise mechanism pathway, surface methyl is formed from protonated methanol firstly and then reacts with toluene to produce protonated xylene and then xylene.

After the formation of xylene in the pores of zeolites as primary products, they need to diffuse out from the catalyst through different channels; however, the diffusion rates of these products in the zeolite are different. In HZSM-5, the diffusion of p-xylene is much faster than that of m-xylene[14,15]; therefore, m-xylene tends to stay in the pores of zeolite for a longer period of time than p-xylene, and further isomerization or alkylation of m-xylene may occur[12]. In this way, diffusion of the primary product formed at the active sites may change the distribution of the final products collected. In addition, the products may further react at the external surface acid sites of zeolite, and the selectivity may be further varied. For example, p-xylene is easily isomerized on the outer surface of zeolites to form m-xylene[12]. One of the challenges in experimental studies is that the information of the primary product distribution cannot be obtained directly.

Deeper understandings of the reaction mechanisms of toluene methylation can be obtained at the molecular level from computational studies, mainly based on density functional theory (DFT) calculations. Bjørnar et al. used DFT to explore the reaction barrier of p-xylene formation from toluene methylation based on the 4T cluster model ($Al(OSiH_3)_3OH$)[16] and found that the barrier was 85 kJ mol$^{-1}$. Recently, Mykela et al. also adopted the method of DFT to calculate the activation barrier of elementary steps in different reaction mechanisms for the generation of three xylene isomers, and they found a co-adsorbate (toluene and methanol) assisted surface methylation mechanism might be preferred at 403 K[17].

In the present study, we performed ab initio molecular dynamics (AIMD) simulations associated with free energy sampling methods to obtain the atomic scale understanding of toluene methylation with methanol in HZSM-5. The AIMD methods would go beyond the static DFT calculations to capture the dynamic nature of chemical reactions under realistic conditions and the flexibility of the zeolite framework in a much more efficient way, especially in the studies of confined or high-temperature systems. The steps of toluene methylation to protonated-xylene, deprotonation of protonated-xylenes, and diffusion of xylene in HZSM-5 channels were considered. The distribution of p-xylene and m-xylene as primary products and the impact of diffusion were investigated by examining the transition state (TS) structures and free energy profiles in each reaction step. Based on these results, the origin of the selectivities was discussed.

## Results

**Reaction pathways and the catalyst model.** The considered reaction pathways of toluene methylation with methanol are presented in Fig. 1. These reaction pathways contain several steps including, (i) reaction between methanol and toluene at Brønsted acid site of HZSM-5 to form proton–PX ($R_1$) and proton–MX ($R_2$), (ii) isomerization between proton–PX and proton–MX ($R_3$ and $R_{-3}$), (iii) deprotonation of proton–PX and proton–MX to form p-xylene ($R_4$) and m-xylene ($R_5$), and (iv) p-xylene and m-xylene migrate along the channels in HZSM-5 through diffusion

**Fig. 1 Reaction pathways of toluene methylation in the pores of HZSM-5.** The steps considered here include the methylation of toluene to proton–PX ($R_1$) and proton–MX ($R_2$), the isomerization between proton–PX and proton–MX ($R_3$ and $R_{-3}$), the deprotonation of proton–PX and proton–MX to p-xylene ($R_4$) and m-xylene ($R_5$), respectively, and the diffusion of these two products ($R_6$ and $R_7$).

($R_6$ and $R_7$). It should be mentioned that for step (i), we considered the direct and the stepwise mechanisms first (details in Supplementary Note 1) and found that the direct mechanism should be favored, which is consistent with the literature results[17–19].

The catalyst HZSM-5 was modeled in a periodic orthorhombic cell. The cell parameters are $a = 20.201$ Å, $b = 19.991$ Å, $c = 13.469$ Å, $\alpha = \beta = \gamma = 90°$, which were taken from literature[20], and a 50 ps NPT simulation was performed to confirm that the cell parameters used are reliable (details in Supplementary Note 2). A substituted aluminum atom was placed at the T12 site widely considered in the literature[21,22], creating a Brønsted acid site at the channel intersection where enough space could be ensured for the reaction to happen (see Fig. 2a). For clarity, we have labeled the carbon atom in methanol as $C_{me}$, the oxygen and hydrogen atoms of hydroxyl in methanol as $O_{me}$ and $H_{me}$, respectively, the oxygen and hydrogen atoms at the Brønsted acid site within zeolite as $O_z$ and $H_z$, respectively, the p-carbon, m-carbon, and p-hydrogen atoms as $C_p$, $C_m$, and $H_p$, respectively, as one can find from Fig. 2b.

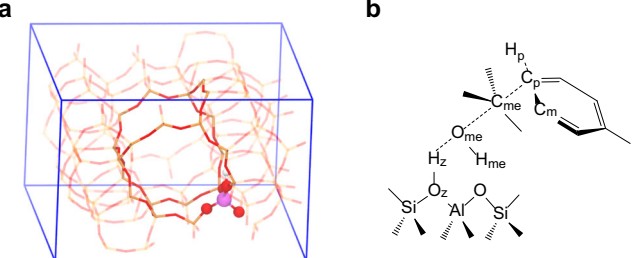

**Fig. 2 The catalyst models applied in the simulations. a** Structure of the periodic 96 T HZSM-5 model; **b** illustration of the toluene direct methylation reaction at the Brønsted acid site. Atom labels as referred in the text and will be used throughout the paper.

**Free energy of toluene methylation to protonated xylene.** Following the preferred direct mechanism in the process of proton–PX formation from methanol and toluene, the $C_{me}$–$O_{me}$ bond is dissociated and $C_{me}$–$C_p$ bond is formed. In Fig. 3, a two-dimensional (2D) free energy surface of toluene methylation to form proton–PX was constructed with metadynamics (MTD) simulation, namely MTD–PX simulation. We can clearly identify the reactant, TS, and intermediate regions in Fig. 3 along the reaction path. In this figure, one initial state (IS) and three intermediate (proton–PX, proton–MX, and proton–OX) basins were marked with green dots. The IS basin represents the state of methanol adsorbing at the $H_z$ of Brønsted acid site, and toluene locating in the channel intersection. Two saddle points $TS(R_1)$ and $TS(R_3)$, corresponding to the TS of toluene methylation and TS of isomerization between proton–PX and proton–MX, were identified and marked with blue dots on the free energy surface. The selected conformations representing those structures in different regions were also shown in Fig. 3.

It is interesting to find from Fig. 3 that the formation of these isomers are sharing a common saddle point, and the barriers to form proton–PX and proton–MX are almost identical. We will show later that such negligible difference between TS energies is originated from the similar TS structures obtained. In addition, one can find from Fig. 3 that the energy of $TS(R_1)$ is higher than that of $TS(R_3)$, and therefore $TS(R_1)$ should be the rate-controlling TS in the initial methylation period, indicating that the formation rates of proton–PX, proton–MX and proton–OX would be similar in this period.

A one-dimensional profile was further constructed by projecting the two-dimensional free energy surface onto diagonal CV1–CV2[22,23]. The averaged 1D free energy profiles were obtained based on the method proposed by Bussi et al.[24], (see details in Supplementary Note 3). Specifically, Supplementary Fig. 6 shows that free energy barriers of the methylation reaction from IS to $TS(R_1)$ ($\Delta A^{\ddagger}(R_1)$) and isomerization from proton–PX to $TS(R_3)$ ($\Delta A^{\ddagger}(R_3)$) are 82.4 and 37.6 kJ mol$^{-1}$, respectively, and the reaction free energy from IS to proton–PX

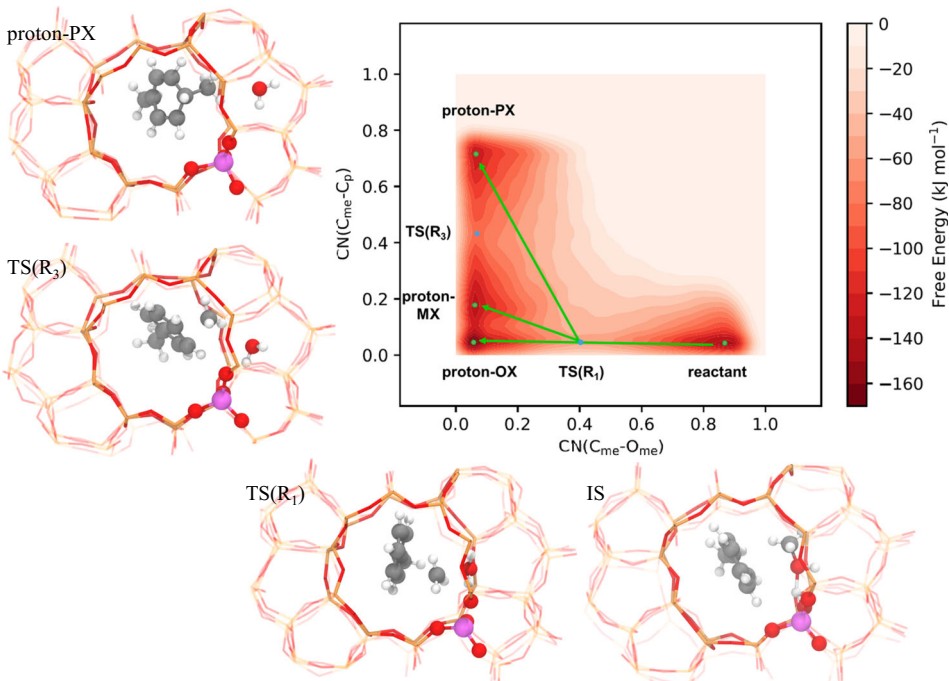

**Fig. 3 Two-dimensional free energy profile of the methylation of toluene.** The profile was obtained from metadynamics simulations. The selected conformations in this process are also presented. (Color code: gray, C; red, O; white, H; orange, Si; and pink, Al).

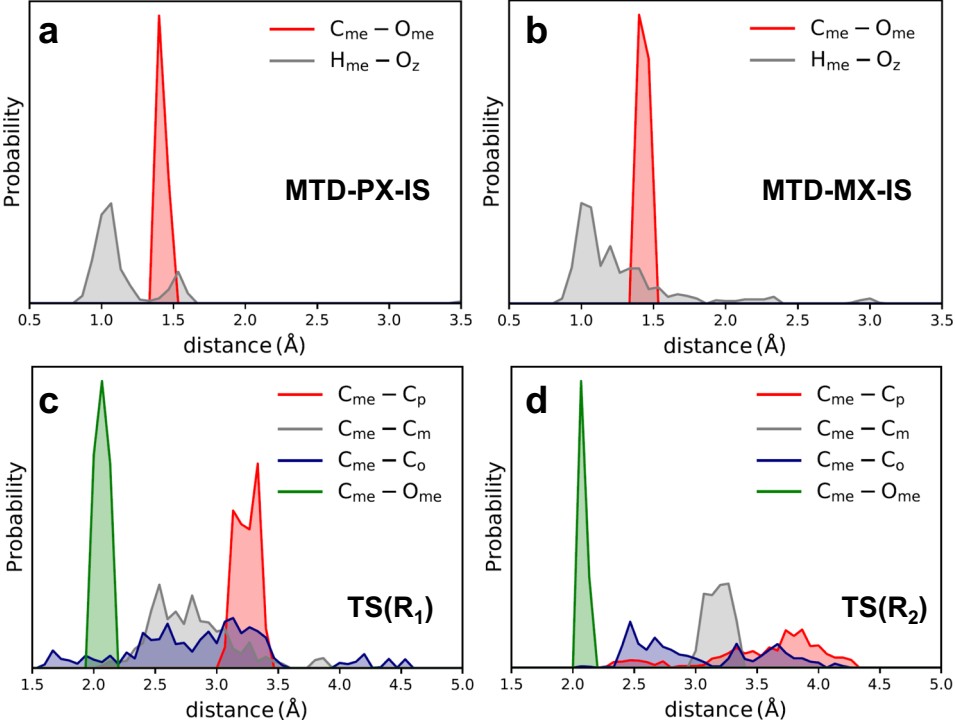

**Fig. 4 Structural analysis of IS and TS regions.** Frequency distribution of distance between atoms in **a** MTD-PX-IS, **b** MTD-MX-IS, **c** TS(R$_1$), and **d** TS(R$_2$) regions in MTD–PX and MTD–MX simulations.

($\Delta A(R_1)$) is 6.3 kJ mol$^{-1}$. However, the basins of the proton–MX and proton–OX shown in Supplementary Fig. 6 are slightly overlapped, so the free energy for proton–MX and proton–OX in the one dimensional (1D) profile are not reliable (see details in Supplementary Fig. 7).

In order to distinguish the proton-MX basin from the overlapping area, we further selected the CN of C$_{me}$–O$_{me}$, and C$_{me}$–C$_m$ as CV1 and CV2, and performed another MTD simulation on the reaction of proton–MX (MTD–MX) generation from methanol and toluene. The corresponding 2D free energy surface was plotted in Supplementary Fig. 8, which is similar to the results in the MTD–PX simulation as shown in Fig. 3. The activation free energy of methylation ($\Delta A^{\ddagger}(R_2)$) and isomerization ($\Delta A^{\ddagger}(R_{-3})$) was calculated to be 88.8 and 34.3 kJ mol$^{-1}$, respectively, and the reaction free energy ($\Delta A(R_2)$) was 13.9 kJ mol$^{-1}$. One can find that $\Delta A^{\ddagger}(R_2)$ obtained here is close to the value of $\Delta A^{\ddagger}(R_1)$ determined in the MTD–PX simulation.

It should be noted that the free energy in standard (i.e., non-well-tempered) MTD simulation is a cumulative bias potential, and there will always be some fluctuations on the energies calculated[25,26]. Therefore, we further evaluated the accuracy of calculated free energy barriers and reaction free energies. By analyzing the changes of $\Delta A^{\ddagger}(R_1)$, $\Delta A^{\ddagger}(R_3)$, and $\Delta A(R_1)$ overtime in the last 100 ps of MTD–PX simulation, we can determine the moment that the free energy minima are filled with Gaussian hills ($t_{fill}$). Based on the same method used above[24], we obtained the averaged 1D free energy profile after $t_{fill}$ and the corresponding error bar (details are provided in Supplementary Note 3). Supplementary Figure 6 shows that, in the MTD–PX simulations, the variation of $\Delta A^{\ddagger}(R_1)$, $\Delta A^{\ddagger}(R_3)$, and $\Delta A(R_1)$ are all within 10 kJ mol$^{-1}$, and in the MTD–MX simulations, the variation of $\Delta A^{\ddagger}(R_2)$, $\Delta A^{\ddagger}(R_{-3})$, and $\Delta A^{\ddagger}(R_2)$ are also within 10 kJ mol$^{-1}$. These results suggest that the MTD simulations performed show adequate accuracy for free energy calculations. In the following discussions, the free energy changes were taken from the averaged 1D free energy profiles.

**Structural analysis of the MTD simulation results**. The structures of different states within the MTD–PX and MTD–MX simulations were further analyzed. Firstly, a series of configurations locating in the IS basins on the 2D free energy surfaces were selected (more details are provided in Supplementary Table 1). The distance between C$_{me}$ and O$_{me}$ atoms and the distance between H$_{me}$ and O$_z$ atoms were summarized in Fig. 4a, b. The quadratic walls added for H$_z$–O$_{me}$ and H$_{me}$–O$_{me}$ were the same to ensure that methanol is in a protonated state (Supplementary Table 4), and H$_z$ is equivalent to H$_{me}$ in this simulation. Moreover, considering the four O$_z$ atoms linked to Al atom are approximately equivalent, only the shortest one of these eight H$_{me}$–O$_z$ bonds in each structure was analyzed. Figure 4a shows that in the IS basin of MTD–PX simulation, the bond length of C$_{me}$–O$_{me}$ is 1.5 Å and the distance of H$_{me}$–O$_z$ is mostly shorter than 1.5 Å, indicating the IS of this reaction can be regarded as the co-adsorption of toluene and methanol at Brønsted acid sites. Similar results were obtained from the MTD–MX simulation, as shown in Fig. 4b. Therefore, one can find that the IS of the two MTD simulations is the same.

As for TS(R$_1$) and TS(R$_2$) regions, the distances of C$_{me}$–C$_p$, C$_{me}$–C$_m$, C$_{me}$–C$_o$, and C$_{me}$–O$_{me}$ were analyzed, the results of which are shown in Fig. 4c, d. Since there are two equivalent $m$- and $o$-carbon atoms in toluene, the shorter one between C$_{me}$–C$_m$ and C$_{me}$–C$_o$ were summarized and analyzed. From the frequency distribution diagram and Supplementary Table 3, the distance of C$_{me}$–O$_{me}$ was close to 2.1 Å, and the average distances of C$_{me}$–O$_{me}$ and C$_{me}$–C$_{p/m/o}$ are quite similar in two MTD simulations, suggesting that the structures of TS(R$_1$) and TS(R$_2$) are quite similar. Both MTD–PX and MTD–MX simulations showed similar TS structures and activation free energies of methylation, demonstrating the robustness and reliability of the calculations used in these studies. The structural analysis of the distance of C$_{me}$–C$_p$, C$_{me}$–C$_m$, and C$_{me}$–C$_o$ in MTD–PX and MTD–MX simulations confirmed the production of desired intermediates, i.e., proton–PX and proton–MX (Supplementary

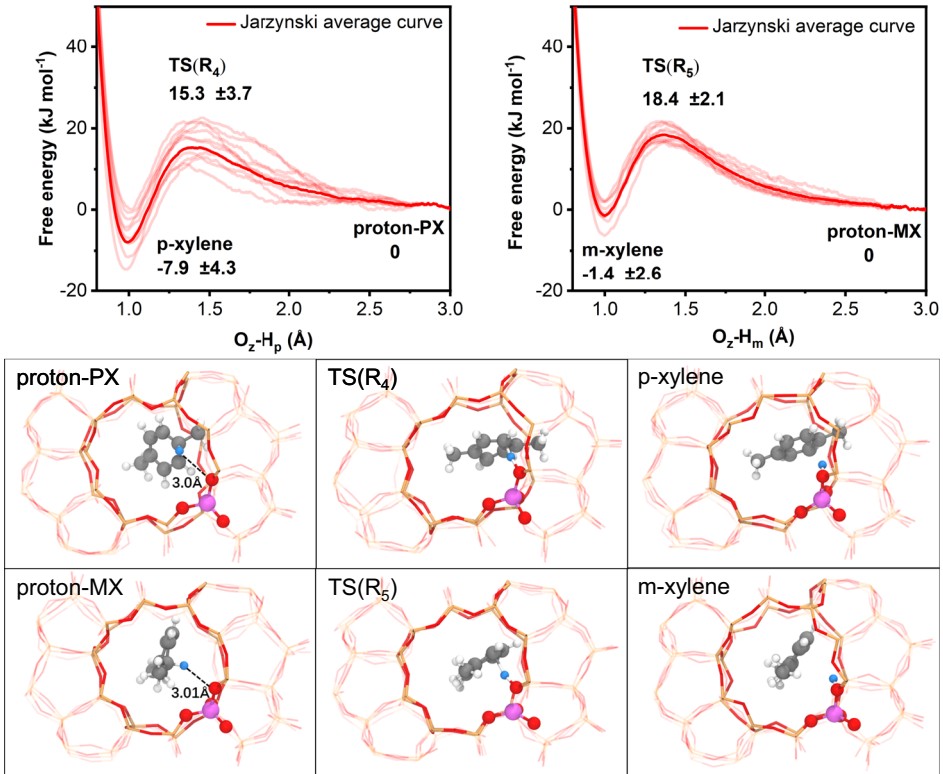

**Fig. 5 Free energy profiles of proton–PX and proton–MX deprotonation.** These profiles were obtained from slow-growth simulations. Thin lines are the work curves of individual simulations and the thick line is the free energy profile obtained from Jarzynski equality. The values of free energy and the standard deviation of works are also shown in the figure. The corresponding structures of the reactants, transition states and products for p-xylene and m-xylene are presented in the lower panel. The blue ball represents the reacting H atom.

Fig. 9). It should be noted that the analysis is done on biased simulation trajectories and the number of structures in Supplementary Tables 1 and 2 can only be used to demonstrate that the selected structures were adequate, but do not represent the equilibrium properties.

**Deprotonation of protonated xylenes.** Figure 5 shows free energy profiles of the deprotonation of proton–PX and proton–MX, and the corresponding selected configurations of the reactants, TS and products. The free energy profiles were obtained from ten parallel slow-growth simulations and calculated with Eq. 2. One can find from Fig. 5 that the activation free energy and reaction free energy are both small for the deprotonation of proton–PX and proton–MX, indicating that such reactions can take place readily under the reaction conditions considered here. Here, 3.0 Å of $O_z$–$H_{p/m}$ distance was considered as the initial reaction coordinate because at such distance, $O_z$ and $H_{p/m}$ are far enough where there are only weak van der Waals interactions between them and it can represent the free energy minimum. Further simulations were performed to confirm this in Supplementary Note 8.

**Summary of primary product formation at the acid site.** The free energy profiles of primary product formation at the Brønsted acid site, including the processes of methylation ($R_1$, $R_2$), isomerization ($R_{3/-3}$) and deprotonation ($R_4$, $R_5$), were summarized in Fig. 6. One can find that the activation free energy of toluene methylation to form proton–PX ($\Delta A^{\ddagger}(R_1)$, 82.4 kJ mol$^{-1}$) is much higher than that of proton–PX isomerization ($\Delta A^{\ddagger}(R_3)$, 37.6 kJ mol$^{-1}$) and proton–PX deprotonation ($\Delta A^{\ddagger}(R_4)$, 15.3 kJ mol$^{-1}$). Similarly, the activation free energy of toluene methylation to form proton–MX ($\Delta A^{\ddagger}(R_2)$, 88.8 kJ mol$^{-1}$) is much

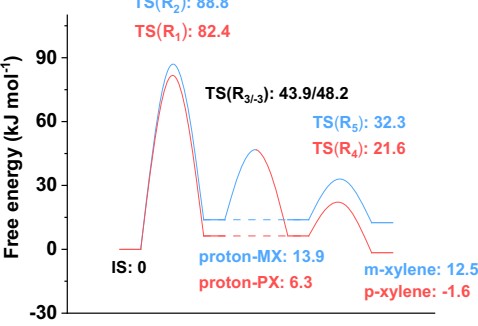

**Fig. 6 Free energy profile of the reactions between toluene and methanol.** The reactions considered include the formation of p-xylene and m-xylene, and the mutual isomerization of p-xylene and m-xylene. The states linked by dashed lines indicate the same intermediates in the reaction.

higher than that of proton–MX isomerization reaction ($\Delta A^{\ddagger}$ ($R_{-3}$), 34.3 kJ mol$^{-1}$) and proton–MX deprotonation reaction ($\Delta A^{\ddagger}(R_5)$, 18.4 kJ mol$^{-1}$). It is obvious that methylation has the highest free energy barrier among all the reactions, and therefore can be considered as the rate-controlling step. The effective free energy barrier of the whole reaction of toluene to p-xylene is 82.4 kJ mol$^{-1}$, which lies within the range of activation energies reported in the literature[10–12,17,27–29].

The effective barriers of the isomerization reaction between p-xylene and m-xylene, following the pathway of p-xylene ↔ proton–PX ↔ proton–MX ↔ m-xylene, can be obtained (see Fig. 6), which are 45.5 kJ mol$^{-1}$ (p-xylene to m-xylene) and 35.7 kJ mol$^{-1}$ (m-xylene to p-xylene). These effective free energy barrier values are close to experimental results of 31.5–43.7 kJ mol$^{-1}$ (m-xylene to

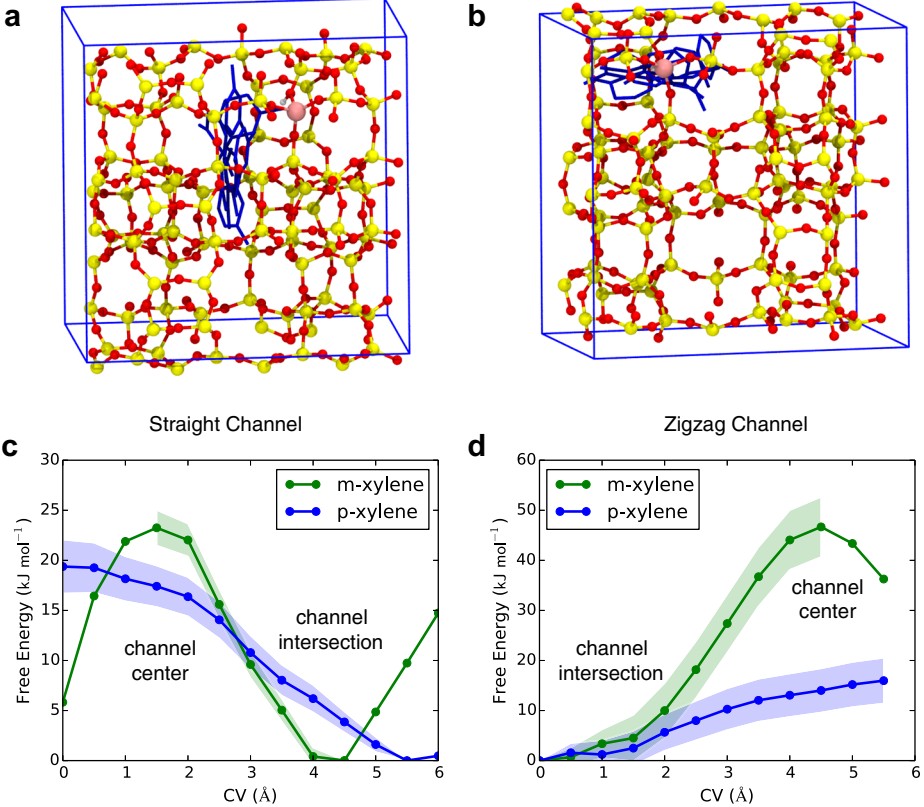

**Fig. 7 The diffusion path and free energy profiles of xylenes in HZSM-5. a, b** Possible paths for p-xylene diffusion at the intersection tunnels. Free energy profiles with a standard deviation of p-xylene and m-xylene diffusing **c** along straight channel; **d** along zigzag channel obtained from blue-moon simulations.

p-xylene)[1,12,30]. The free energy barriers of (de)protonation reactions are much lower compared with those of methylation and isomerization. In addition, Fig. 6 shows that the free energy barrier of deprotonation ($\Delta A^{\ddagger}(R_4)$ or $\Delta A^{\ddagger}(R_5)$) is much lower than that of the corresponding isomerization ($\Delta A^{\ddagger}(R_3)$ or $\Delta A^{\ddagger}(R_{-3})$), indicating that deprotonation can happen readily upon the formation of proton–PX or proton–MX. From Fig. 6, we can also see that p-xylene is 14.1 kJ mol$^{-1}$ more stable than m-xylene in HZSM-5. Therefore, it is obvious that the effective barriers for the formation of p-xylene and m-xylene are almost identical, whilst p-xylene is slightly more stable than m-xylene in the catalyst pores.

**Diffusion of xylene in HZSM-5 channels**. Upon the formation of p-xylene and m-xylene molecules at the active sites within HZSM-5, there are two competing processes, namely diffusion out of the zeolite, and the isomerization between p-xylene and m-xylene. Therefore, we further calculated the diffusion barriers for p-xylene and m-xylene from one channel intersection to the adjacent and compared these barriers with the isomerization barriers.

There are two types of channel systems in HZSM-5, namely straight channels, and zigzag channels (see Fig. 7a, b). Here, the free energy barriers of xylene diffusion across the channel intersections through these channels were calculated. Considering the translational symmetry of HZSM-5, only the diffusion path from the intersection along one channel to the center of the channel was evaluated. It was shown in Fig. 7c, d and Supplementary Table 5 that the diffusion barriers for p-xylene along straight channels ($\Delta A^{\ddagger}(R_{6-s})$) and zigzag ($\Delta A^{\ddagger}(R_{6-z})$) channels are 19.4 and 15.9 kJ mol$^{-1}$, respectively, whilst for m-

xylene, the corresponding barriers are 23.2 kJ mol$^{-1}$ ($\Delta A^{\ddagger}(R_{7-s})$) and 46.8 kJ mol$^{-1}$ ($\Delta A^{\ddagger}(R_{7-z})$), respectively. Previous experimental results showed that at the same conditions, the diffusion of p-xylene is always faster than m-xylene, which is consistent with our results[14,15,31–33].

## Discussion

With all the results obtained above, a simplified mechanistic framework combining the processes of isomerization and diffusion can be obtained, which is shown in Fig. 8. For p-xylene, we find that the isomerization showed a much higher barrier than diffusion in HZSM-5, thus it tends to diffuse away from zeolite rather than undergo isomerization. In contrast, the diffusion barrier of m-xylene along the zigzag channel is much higher than the corresponding isomerization barrier. Therefore, one can find that m-xylene prefers to undergo isomerization into p-xylene.

We noticed that, in a recent experiment, Wang et al. improved the selectivity of p-xylene significantly through an approach of increasing the ratio of sinusoidal pore openings in the HZSM-5-catalyst[7]. The experimental results are consistent with the results obtained in the current work. As one can find from Fig. 8, $\Delta A^{\ddagger}$ ($R_{7-z}$) is higher than the isomerization barrier of m-xylene. When m-xylene can only escape from zeolite through the zigzag channel, isomerization of m-xylene to form p-xylene is preferred and therefore the selectivity of p-xylene is increased.

Summing up, we showed that the methylation reaction of toluene and methanol in HZSM-5 to form p-xylene and m-xylene through similar TS and activation barrier, thus p-xylene and m-xylene are generated with a similar possibility in the primary product. Methylation is the rate-determining step of the whole reaction with the highest free energy barrier. Isomerization can

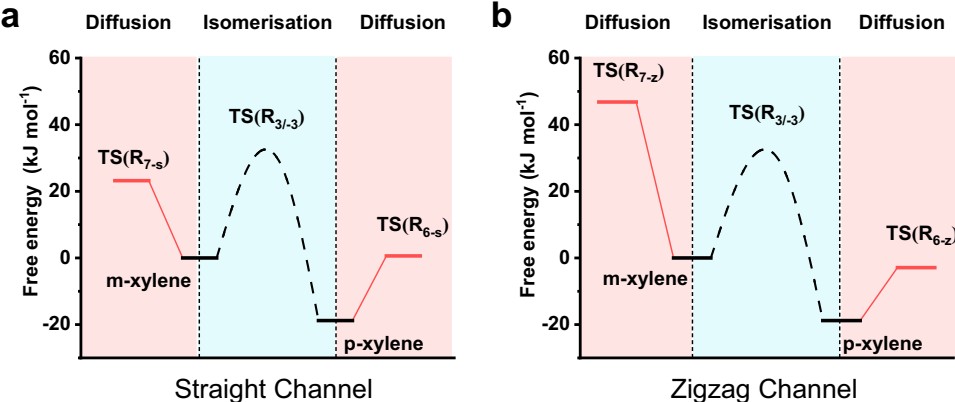

**Fig. 8 Competition between diffusion and isomerization of xylenes in HZSM-5.** The simplified mechanistic framework of toluene methylation to p-xylene and m-xylene by combing the isomerization and diffusion along **a** straight and **b** zigzag channels.

occur between p-xylene and m-xylene through the protonated intermediates, and the effective isomerization barriers are only around 40 kJ mol$^{-1}$. In addition, p-xylene is slightly more stable than m-xylene in HZSM-5, which promotes the selectivity of p-xylene formation in HZSM-5. More importantly, the diffusion barrier for p-xylene is lower than m-xylene in HZSM-5, and the diffusion barrier of m-xylene along the zigzag channel is higher than the barrier of its isomerization to p-xylene, which further promotes the selectivity of p-xylene formation in HZSM-5. This work made a direct comparison for the first time between all the possible processes in toluene methylation at the molecular level, based on simulations all at DFT-level. These insights obtained are crucial for the further development of high-performance zeolite catalysts for toluene methylation.

## Methods

**AIMD simulation.** AIMD simulations were performed at 670 K in a constant volume and constant temperature (NVT) ensemble using CP2K simulation package[34] with Gaussian plane-wave basis sets (GAW). The revPBE functional was employed to set the plane-wave basis due to its improved ability to describe energies related to catalytic processes compared with the commonly used functionals, e.g., PBE, for solid-state calculations[35,36]. The Goedecker–Teter–Hutter (GTH) norm-conserved pseudopotentials were used for the description of the core electrons, and the shorter range molecularly optimized GTH basis sets[37] with an energy cutoff of 280 Ry was chosen to expand electronic wavefunctions. DFT-D3 method was utilized to better describe dispersive interactions[38–40]. The time-step used in the MD simulations was 0.5 fs. Nosé–Hoover thermostat with three chains was used to maintain the temperature at 670 K[41,42].

**Free energy sampling.** The direct mechanism of toluene methylation mainly consists of two sequential processes. The initial process is the addition of methyl group to toluene producing protonated xylene, which was simulated with the MTD method. This is because the reaction considered contains serval metastable states and TS, and the MTD method could explore high-dimensional free energy surfaces to distinguish these states[26,43]. The second process is the deprotonation of the protonated xylene to generate xylene molecule, and this reaction is relatively simple and therefore the more efficient slow-growth approach[44–46] was used to study this process. Subsequently, the diffusion process was modeled using constrained MD based on the blue moon sampling method.

**MTD simulations.** In the MTD simulations, the coordination number (CN) between two reacting atoms was used as the collective variable (CV). The CNs were defined by:

$$\text{CN} = \sum_{i,j} \frac{1 - \left(r_{ij}/r_0\right)^{nn}}{1 - \left(r_{ij}/r_0\right)^{nd}} \quad (1)$$

where $r_{ij}$ was the distance between atoms $i$ and $j$, $nn$, and $nd$ were set to 6 and 12, respectively, and $r_0$ was set to 2.0 Å in order to differentiate the regions of reactants, products, and TS in the current work.

In the MTD simulation of proton–PX formation (MTD–PX), the CN between $C_{me}$–$O_{me}$ in methanol was selected as CV1, the CN between $C_p$–$C_{me}$ was selected as CV2. While for the simulation of proton–MX formation process (MTD–MX), CV1 was still the CN between $C_{me}$–$O_{me}$, CV2 was the CN between $C_m$–$C_{me}$. The height of Gaussian hills used in the MTD simulations was 2.6255 kJ mol$^{-1}$ (0.001 Hartree) initially, then reduced to 1.3128 and 0.6564 kJ mol$^{-1}$ to improve the accuracy of the results. The width of Gaussian hills was 0.03 and the hill's spawning time step was 25 fs (50 steps). In addition, quadratic walls were used to restrict the CVs on the regions of free energy surface of interest (see Supplementary Table 4). The trajectories of CV1–CV2 and the moments to reduce the height of Gaussian hills in the MTD–PX/MTD–MX simulations are shown in Supplementary Fig. 13.

Then by taking the difference between CVs, i.e., CV1–CV2, as the reaction coordinate, we projected the 2D free energy surface onto 1D (the sampling density on the projection axis was 0.01) to further obtain the 1D free energy profiles. The methylation barrier, isomerization barrier, and reaction-free energy were obtained from the 1D free energy profiles. The projecting method was introduced in the work of Moors et al.[22], with the following equation

$$G(\text{CV1} - \text{CV2}) = -\frac{1}{\beta}\ln\left\{\int_{-\infty}^{\infty} d_{\text{CV1}} \exp\left[-\beta G(\text{CV1} - \text{CV2, CV1})\right]\right\} \quad (2)$$

**Slow-growth simulations.** For the subsequent deprotonation of proton–PX, a slow-growth approach was used in the AIMD simulations. We first selected the representative configuration in the MTD product area with the water molecule removed as the input, then equilibrated the system for 5 ps. Taking the distance of $H_p$–$O_z$ as the reaction coordinate, the distances were changed slowly with a rate of −0.000109 Å frame$^{-1}$ from the initial distance of 3.0 Å. Then the free energy profile of the deprotonation of proton–PX can be obtained. Considering the possible errors in one free energy profile, ten parallel simulations were conducted with the same parameter settings (see Supplementary Note 8), then the Jarzynski equality was used to calculate the average result[47]

$$\triangle A = -\beta^{-1}\overline{\exp(-\beta W)} \quad (3)$$

where $\triangle A$ is the Helmholtz free energy difference, $\beta = 1/k_BT$, $W$ is the work performed on the system. Similarly, for the deprotonation of proton–MX, we took the distance between the meta-hydrogen ($H_m$) in proton–MX and $O_z$ as the reaction coordinate, and all other parameter settings were the same.

**Blue moon sampling simulations.** Free energy samplings on the diffusion of xylene in zeolite channels were performed with the "blue moon" sampling method and constrained MD simulations. The HZSM-5 model was also employed in the simulation. The distance $l$ between the center of xylenes and the center of the zeolite along the $x$-axis (zigzag channel) or $y$-axis (straight channel) were defined as CVs. The constraint force $F_c$ required to constrain the reaction coordinates at each $l$-value was evaluated as a function of $l$ from individual AIMD trajectories. To determine the diffusion free energy profiles of m-xylene and p-xylene through ZSM-5 channels, the potential of mean force was calculated by integrating $\langle F_c \rangle$ over the reaction coordinate

$$\triangle A = \int \langle F_c \rangle dl \quad (4)$$

Each simulation was performed with 5000 steps. The first 1000 steps at the beginning of the constrained MD simulations were omitted and the constraint force was collected from the last 4000 steps (see details Supplementary Fig. 12).

## Data availability
The data supporting this study are available from the corresponding author upon reasonable request.

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

## Acknowledgements
This work was supported by the Shanghai Rising-Star Program (20QA1406800), the National Natural Science Foundation of China (22072091), and ShanghaiTech University. We thank the HPC Platform of ShanghaiTech University and Shanghai Supercomputer Center for computing time.

## Author contributions
Q.C. performed the AIMD simulations related to methylation, deprotonation, and isomerization. J.L. performed the simulations related to diffusion. J.L. and B.Y. conceived the problem. All the authors contributed to writing the paper.

## Competing interests
The authors declare no competing interests.
