## [Peer Review File · Nature Communications]

Editorial Note: In their review of the first version of this manuscript, Reviewer #2 added some comments to the manuscript file. These comments, excluding minor textual revisions, have been copied into this Peer Review File on page 3.

REVIEWER COMMENTS

Reviewer #1 (Remarks to the Author):

This manuscript by Yang and co-workers report a very interesting study on the selectivity of zeolite catalyzed toluene methylation. As stated by the authors, this is a very important reaction with many features that are not well understood nowadays. The manuscript is well structured and the computational method applied by the authors definitely goes beyond the state of the art. However, I have quite some concerns that prevent me from advising to publish the manuscript in its current form in Nature Communications. Generally speaking, the detailed way the manuscript is written, does not appeal to the broad readership of this journal. Moreover, the current manuscript does not provide ground breaking new insights and as explained below, I have some doubt about the quality of some of the results. Provided the necessary modifications are made, I judge that this manuscript would be better suited for a more specialized (catalysis) journal. Below I list my major concerns:

- The authors opted for a dynamic approach. To appreciate the enormous effort that goes into such approach, it should be described why the authors expect a static approach to fail in this case.
- It is well known that the competition between the stepwise and concerted methylation reaction can be driven by parameters such as partial pressures and temperature. To study this, typically a microkinetic model would be needed. In this work, the authors just make an a priori choice without explicitly studying this effect. Moreover, the choice for the direct reaction pathway is only motivated by some references with some of them being more than 10 years old. Are the conditions mimicked in those studies relevant for the ones looked at in this study? Do the conclusions of this study still hold at higher methanol loadings?
- As is commonly done, the authors study the T12 Bronsted acid site in ZSM-5. It would, however, be interesting to verify to what extent the conclusions depend on parameters such as the location of the acid site and acid site density. I realize that it easily becomes infeasible to compute all possible combinations of acid sites, but one can easily imagine that shape selectivity in a complex cavity structure like ZSM-5 might be different for different T sites.
- The methods section describes that the unit cell was taken from the zeolite database. This implies that the periodic unit cell used in the entire study was not relaxed or equilibrated at the temperature used in this study. It is known that zeolites exhibit negative thermal expansion and therefore it is crucial to work with a relaxed or equilibrated unit cell before performing production runs with MD.
- Details like the applied Gaussian basissets (DZVP, TZVP ...) for the GPW approach or the hills spawning time for the metadynamics simulations are lacking.
- Were the production runs stable enough with just a single Nosé-Hoover thermostat? As can be seen in literature, it is common to use a chain of NH thermostats.
- How were free energies calculated from metadynamics runs?
- How did the authors choose the collective variables for the free energy sampling? It also seems that throughout the studies various types of collective variables have been mixed (coordination numbers and distances). Did the authors check whether these are the appropriate CVs?
- I am not convinced that all reported free energies are fully converged. Therefore it is needed to report error bars. Figure S3 for example does not convincingly show that the numbers are converged (very sudden jumps etc.). What is the free energy spike in the intermediate structure in figure S1?
- Why were various types of free energy sampling methods used? Using all methods for all reactions could also give a feeling of how reliable the results are. Now the choice is not motivated and seems to be rather random. Maybe the authors had some practical reasons for this choice?
- The free energy curves in Figure 4 do not show a clear free energy minimum for proton-PX and proton-MX. Was the sampling sufficient? Maybe sampling at larger distances was needed as well?
- The authors state "The effective barrier of the whole reaction of toluene to p-xylene is 82.2 kJ/mol, which is close to the values of 60.52 to 86.41 kJ/mol obtained in pervious experiments and simulations". If the authors call a difference of roughly 22 kJ/mol close, what is then considered as accurate?
- Based on Figure 5, the authors conclude that the probability to form p-xylene and m-xylene is nearly identical. From a kinetic perspective this is true, however there is a difference in reaction free energy, so from a thermodynamic viewpoint one would expect a difference (preference for p-xylene).

Reviewer #2 (Remarks to the Author):

The manuscript of Chen, Liu and Yang investigates the key steps in selectivity for toluene methylation, and concludes that m-xylene will preferentially isomerise to p-xylene, and then diffuse out of the catalyst zeolite, because the m-xylene cannot diffuse as easily. The claims are novel and would be interesting to a wide audience, because they offer atomistic insight into the mechanism in an industrially valuable process.

The work is convincing, though this reviewer would like to see several questions addressed for the scientific conclusions to be rigorous:

- with respect to the previous literature, the authors change their notation for the catalyst from ZSM-5 to H-ZSM-5 after paragraph 2. Is this deliberate? Have other counterions been considered previously? It would be good to clarify this point.
- The authors might want to explain why they chose to run simulations at 670 K, given experiment demonstrates the direct mechanism occurs at very slightly higher (> 673 K; is this industrial conditions?). Why not at higher temperatures? Presumably the stepwise mechanism is competitive at the chosen temperature - it would have been informative to compare both.
- What temperature were the crystal structure parameters obtained for? And more importantly, what cell parameters do the simulation settings return with e.g. NPT? How much discrepancy is there between the used and equilibrium lattice constants, and does this introduce an error through e.g. lattice strain?
- Why was T12 chosen for the aluminium site? How does this influence the results? And why was this Si/Al ratio chosen?
- For Figure 2, some clarity would be good as to why formation rates of protonated xylenes would be equivalent; reviewing the diagram, the pathway seems to go through proton-OX and proton-MX, which would therefore have greater rates? This also appears the case from the diagonal CV2-CV1 plot (L138), where the OX and MX intermediates are lower in energy?
- On L195, the TS structures are noted as being similar but C_o differs significantly. Why is this? Also close by, it might be nice to comment on the intermediate configurations in the pore (obvious from Fig S4)
- On L215, discussion is given of proton migration. Can the authors confirm this is barrierless? As it isn't clear from the text and diagrams.
- For the diffusion studies, convenience is not an appropriate justification for use of a silicate model. In reality, the aluminium are present and so here the diffusion is artificially elevated; what would be the consequence of having aluminium sites on the diffusivity, and does this alter the conclusions?
- On L265, diffusion rates are eluded to. This should allow derivation of hopping rates that can be compared to the authors results, assuming there is a hopping mechanism?

There are also aspects of the methodology and statistical analysis that the reviewer thinks could be clarified:

L64: Please define T for 4T. What is the difference between this and T12 also needs explaining briefly.

L171: "Series of configurations" - how many, and selected by what justification? Are there error bars from the sampling?

L193: "C_m is much higher than ..." - can this be made quantitative with e.g. centre of mass differences?

Fig6: Are there error bars for the free energy landscapes?

L315: Why did the authors chose the exchange-correlation functional, dispersion corrections, and timestep chosen for the MD?

L334: How was the value of r₀ chosen?

L356: Naive question, but Eq2 doesn't show obviously how the averaging is achieved - no summation?

L369: "I-value" doesn't have a definition of "I"?

L375: Why the last 4 ps of the trajectory?

Finally, minor spelling/grammatical mistakes, clarifications/corrections/improvements to images, and inconsistencies in the bibliography presentation are included in the attached PDF as comments.

Page 4, Scheme 1: "Figure needs to show charge on Bronsted acid and base pairs."

Page 9, Fig. 4. "Improve quality of x-axis as cannot read angstrom sign?. y-axis definition of kJ/mol different from previously?"

Page 15 second row: suggested amendment of "The initial process is the addition of methyl group..." to "The initial process is the addition of a proton."

Reviewer #3 (Remarks to the Author):

The authors present a computational study into the Methylation of toluene with methanol to produce p-xylene. The authors use ab initio molecular dynamics (AIMD) simulations and well-established free energy sampling methods.

My expertise is in the development and application of enhanced sampling methods so my comments will be mostly focusing on this part of the manuscript.

Overall, the methodology used in the paper is appropriate for tackling the scientific question that the authors want to answer. The simulations seem to be performed properly though some of analysis could be done better, see my comments below.

I believe that the manuscript can eventually be accepted for publication in Nature Communication. However, at the moment, I find various issues with manuscript that need to be addressed before it can be accepted for publication.

I would need to see the manuscript again, but once the authors have addressed the issues raised in my report, I can very likely recommend that that manuscript is accepted.

Please find in the following my comments:

* (Page 1, line 13-14) "and state-of-the-art free energy sampling methods" I would not call the free energy methods used in the paper to be state of the art. Instead they are all well established and proven methods for free energy calculations. Thus, I would ask the authors to change this statement.

* Page 4, line 91 and page 5, line 116) The acronym PX, MX, and OX are never explicitly defined. This should be clarified.

* (Page 7) The authors perform two metadynamics simulations, MTD-PX and MTD-MX. Apart from the statement, "In order to distinguish the proton-MX basin from the overlapping area, we further performed MTD simulations on the reaction of proton-MX (MTD-MX) generation from methanol and toluene.", I find it not well explained the reason for doing the second MTD-MX simulation. I would understand this is due to the choice of the CVs. Since many reader will not be experts on enhanced sampling simulations, they might not understand this issue. Therefore, this should be discussed more clearly. Furthermore, it should be briefly explained in this section how these two simulations differ and what makes the MTD-MX simulation more suited to describe the reaction of proton-MX.

* (Page 7) One issue with standard (i.e. non-well-tempered) metadynamics simulations, like the authors employ, is that it is hard to know when to stop the simulation. This is what the authors discuss on lines 156-167. Therefore, a common solution is to employ a time-averaged estimate of the free energy surface, see Equation 9 in <http://doi.org/10.1038/s42254-020-0153-0>. In this way, one can also assign an error to the free energy estimate. This would be a more appropriate way to analyse the metadynamics simulations and I would strongly encourage the author to perform such analysis. At the very least, they should acknowledge that it is possible to perform such analysis.

* (Page 7) Following at the previous points. For the metadynamics simulations, the authors should assign error bars to their results and values. For example, for Figure S1 and S2 (lower panel) in the SI, it would be good to get a feeling for the error bars.

* (Page 7 lines 156-157) This statement, "It should be noted some fluctuations on the energies calculated." This is strictly only true for standard (i.e. non-well-tempered) metadynamics simulations. The authors should modify the statement to clarify this.

* (Page 7, lines 165-167) I find the following statement to be unclear: "In the following discussion, the free energy differences were taken from the 1D potential energy surface obtained by the sum of all Gaussian hills." The authors employ a 2D metadynamics simulations. Thus from the sum of

the Gaussian hills, they obtain a 2D free energy surface. Then they can perform a 1D projection of the 2D FES. I assume this is what they discussing here, however, I find it badly worded and this should be clarified. Furthermore, what is the collective variable used for the 1D FES? That is not clear. If it is the diagonal CV2-CV1 like mentioned on line 138, this should be mentioned again here.

* Following on the previous point. In multiple places the authors incorrectly use "potential energy surface" when they are referring to a "free energy surface", for example lines 166, 172. The authors need to go through the manuscript and check that this is corrected everywhere.

* (Figure 3, page 8) The frequency distribution for the distances shown in Figure 3 are from metadynamics simulations and thus biased, unless the authors have performed reweighted when generating the distribution. This should be clearly stated in the caption of Figure 3. Furthermore, in the discussion of "Structural analysis of the MTD simulation results", it should be clearly stated that the analysis is done on biased simulation trajectories and thus the numbers obtained do not represent the equilibrium properties. However, it is fine to use the analysis to compare two different metadynamics simulation like the authors do. They should just state more clearly the limitations of the analysis and results.

* (Page 9-10) One issue with using Jarzynski's equality to obtain free energy profiles from slow-growth simulations (i.e, steered-MD) is that one generally needs a large amount of independent simulations to get converged results. Can the authors comments on this since they only perform 5 simulations? Why did they conclude that this is sufficient?

* (Figure 4, page 9) Following on the previous point. In Figure 4, the authors only report the average energy values of the p-xylene and the TS. It would good if they also assign an error bar to the value, which they can obtain by looking at the variance/spread between the 5 different slow-growth simulations.

* (Figure 4, page 9) To make the manuscript clearer, it should be explicitly stated in the caption of Figure 4 that the results are obtained with slow-growth simulations.

* (Page 12, lines 258-261) "...channel is sampled until the maximum free energy, corresponding to the free energy of transition states, was reached." In the blue moon sampling simulations, how do the authors detect the maximum free energy is reached? This is unclear to me and needs to be clarified in the manuscript.

* (Page 12) In the "Diffusion of xylene in ZSM-5 channels" section, the authors should explicitly state that the results are obtained using the blue moon sampling methods. Also in the caption of Figure 6.

* (Page 14) Do the authors have some motivation for using the revPBE functional? Has it been shown to well describe this system? If so, it might be good to mention it in the Methods section on page 14.

* (Page 14) Various details about the setup of the CP2K seem to be missing. For example, the exact type of basis set used (e.g., DZVP-..., etc). Also any cutoffs, if used. Given that there should be enough information to reproduce the results, I ask the authors to add all the relevant information about the setup of the CP2K simulations.

* (Page 14, line 318) The citation for the Nose-Hoover thermostat used in CP2K is missing. This should be added.

* (Page 15) The overall simulation time of the metadynamics simulations is missing. This needs to be added.

* (Page 15) "The height of Gaussian hills used in the MTD simulations was 2.6255 kJ/mol (0.001 hartree) initially, then reduced to 1.3128 and 0.6564 kJ/mol to improve the accuracy of the results." The scheduling policy used here should be explicitly stated, in other words, for how long time was each height value used. For example, for first X ps 2.6255 used, then 1.3128, etc. I

remind the authors again, there should be enough computational details included such that others can try to reproduce the results.

* (Page 15, lines 343-345) "In addition, quadratic walls were used to restrict the CVs on the regions of free energy surface of interest (see Table S1)." I am rather confused by the usage of attractive and repulsive for the direction in Table S1. I assume that the authors mean upper and lower walls that are only active once the CV have reached a certain value to limit the CV space explored by the metadynamics simulations. The usage attractive and repulsive might indicate that the walls are always active. This needs to be clarified. I would recommend the authors to use instead upper/lower for the wall type.

* (Page 15) To clarify, did the authors use the internal metadynamics code in CP2K? Or did they use an external code like PLUMED? If they did use PLUMED, this should be explicitly stated.

* (Page 15-16) The same goes for the slow-growth and blue moon sampling simulations. Did the authors use just CP2K and the free energy methods implemented in the CP2K code? Or did they use any external library/code? Any external library/code apart from CP2K should be mentioned.

This is Omar Valsson, Max Planck Institute for Polymer Research

REVIEWER COMMENTS

Reviewer #1 (Remarks to the Author):

This manuscript by Yang and co-workers report a very interesting study on the selectivity of zeolite catalyzed toluene methylation. As stated by the authors, this is a very important reaction with many features that are not well understood nowadays. The manuscript is well structured and the computational method applied by the authors definitely goes beyond the state of the art. However, I have quite some concerns that prevent me from advising to publish the manuscript in its current form in Nature Communications. Generally speaking, the detailed way the manuscript is written, does not appeal to the broad readership of this journal. Moreover, the current manuscript does not provide ground breaking new insights and as explained below, I have some doubt about the quality of some of the results. Provided the necessary modifications are made, I judge that this manuscript would be better suited for a more specialized (catalysis) journal. Below I list my major concerns:

- The authors opted for a dynamic approach. To appreciate the enormous effort that goes into such approach, it should be described why the authors expect a static approach to fail in this case.

Response: We sincerely thank the reviewer for this suggestion to help us improve the manuscript. The static DFT approaches are widely used in studies of catalytic reactions by many groups including ours. However, these approaches may lead to considerable

errors for the complex system of reacting substances confined in cavities at high temperatures.

Firstly, as the free energy surface of complex systems has several local minima, the static DFT results tend to be strongly dependent on how we set the initial structures for optimization. The structures at finite temperature would be rather different from those obtained at 0 K, and require direct sampling of phase space to resemble the configurational distribution. For example, Bucko *et al.* studied propylene cracking reaction in CHA type of zeolite with a static approach [Bucko, *et al.*, *J. Catal.*, 2011, 279, 220]. Based on a series of different initial structures, they obtained rather distinct transition states and free energy profiles. These results indicate that neither the reactant nor the transition state can be represented with sufficient accuracy by a single configuration determined by a relaxation with static DFT.

Furthermore, the flexibility of the zeolite framework should also be taken into account, because the processes in zeolite catalysts are strongly correlated with the size of molecules and pores in zeolite. For example, some works have shown that higher diffusivities in zeolite catalysts would be obtained when the zeolite framework was treated flexible rather than rigid [e.g. Leroy *et al.*, *Phys. Chem. Chem. Phys.*, 2004, 6, 775; Garcia-Sanchez, *et al.*, *J. Phys. Chem. C.*, 2010, 114, 15068; Kopelevich, *et al.*, *J. Chem. Phys.*, 2001, 115, 9519].

Moreover, in the static approach, all the free energy barriers of surface reactions were estimated by total energy calculations obtained at 0 K with thermodynamic corrections. It should be noted that thermodynamic corrections were built upon some approximations, especially for adsorbates on surfaces, where zero-point energy and entropies were calculated based on vibrational frequencies. Temperature-dependent contributions can be estimated with a harmonic-oscillator or ideal-gas approximation. Those corrections are sufficiently accurate at low temperatures because the potential energy difference makes a dominant contribution to the free-energy barrier, however, the approximations may lead to considerable errors at high temperatures.

We consider that MD simulations would alleviate this drawback and provide useful insights into the understanding of high-temperature reactions in confined space. Following the reviewer's suggestion, we have added in the revised manuscript descriptions explaining why the dynamic approach was used (lines 72-76, page 3).

- It is well known that the competition between the stepwise and concerted methylation reaction can be driven by parameters such as partial pressures and temperature. To study this, typically a microkinetic model would be needed. In this work, the authors just make an a priori choice without explicitly studying this effect. Moreover, the choice for the direct reaction pathway is only motivated by some references with some of them being more than 10 years old. Are the conditions mimicked in those studies relevant for the ones looked at in this study? Do the conclusions of this study still hold at higher methanol loadings?

Response: We thank this reviewer for the suggestions and comments. In recent years, there are lots of work discussing the competition between the two methylation mechanisms. De Wispelaere *et al.* [*Catal. Today*, 2018, 312, 35] explored the competitive relationship of two reaction mechanisms of benzene methylation reaction under different pressures and temperatures through a microkinetic study. They found that as the temperature increases, the stepwise mechanism would prevail.

Following the helpful suggestions from this reviewer, we further performed a series of MTD simulations to explore the free energy profiles of the stepwise mechanism. $C_{me-O_{me}}$ and C_{me-O_z} were selected as CV1 and CV2 to obtain the free energy surface of methyl groups formation at the acid site (the first step of the stepwise mechanism). Figure R1 shows the curve of obtained free energy barrier and reaction free energy against time. We can see that the free energy barrier is greater than 150 kJ/mol (although the fluctuation is relatively large), which is close to the value of 152 kJ/mol reported by Moors *et al.* [ACS Catal., 2013, 3, 2556]. One can see that the free energy barrier to form the surface methyl group is much higher than the effective barrier obtained in the current work for the concerted mechanism, and therefore the concerted mechanism should be preferred under the conditions considered in our work. We have added those results and discussions in the revised manuscript (lines 95-98, page 4 and Supplementary Note 1).

Regarding the issue of methanol loading, there are some literatures focusing on this issue while studying the methylation reaction [For example, Moors, et al., ACS Catal., 2013, 3, 2556; Nastase et al., ACS Catal., 2020, 10, 8904]. It is undeniable that the increase in methanol loading will affect each step of the reaction process. According to the results from Moors *et al.*, as the methanol loading increases, stable protonated methanol clusters will be formed, and this causes the benzene methylation barrier to rise from 118 to 154 kJ mol⁻¹ (1 methanol to 3 methanol cluster). The methylation reaction will still be the rate-determination step. As for the deprotonation reaction, Wispelaere et al. [J. Catal., 2013, 305, 76] reported that both water and methanol can be used as carriers of protons, this will further reduce the barrier of the deprotonation reaction. Therefore, we believe that the increase in methanol loading will not change the conclusions made in the original manuscript.

Figure R1. The curve of free energy barrier and reaction free energy of surface methyl groups formation reaction against time.

- As is commonly done, the authors study the T12 Bronsted acid site in ZSM-5. It would, however, be interesting to verify to what extent the conclusions depend on parameters such as the location of the acid site and acid site density. I realize that it easily becomes infeasible to compute all possible combinations of acid sites, but one can easily imagine that shape selectivity in a complex cavity structure like ZSM-5 might be different for different T sites.

Response: We thank the reviewer for this important comment. The T12 site is located at the intersection of the straight and sinusoidal channels and can provide maximal reaction space for the guest molecules, therefore it was widely considered in previous studies [e.g., Moors, et al., ACS Catal., 2013, 3, 2556; De Wispelaere, et al. Catal. Today, 2018, 312, 35, etc.]. In addition, toluene is a relatively large molecule (7.0 Å in diameter) compared with the channels of ZSM-5 (6.9~7.2 Å in diameter) [Kaeding et al., J. Catal., 1981, 67, 159], and it is reasonable to expect T12 is the most suitable site for the toluene methylation reaction. We fully agree with the reviewer that “shape selectivity in a complex cavity structure like ZSM-5 might be different for different T sites”. It would be absolutely a good topic to be discussed in the future work investigating the role of different sites in the reactions, taking the effects of transition-state-shape selectivity into consideration. Nevertheless, in the current work, we are only considering the T12 site because this site would guarantee that there is enough space for the reaction to happen, thus we can put more focus on the intrinsic reaction kinetics of toluene methylation at easily accessible Brønsted acid sites. The manuscript has been modified to clarify this issue (lines 103-105, pages 4-5).

- The methods section describes that the unit cell was taken from the zeolite database. This implies that the periodic unit cell used in the entire study was not relaxed or equilibrated at the temperature used in this study. It is known that zeolites exhibit negative thermal expansion and therefore it is crucial to work with a relaxed or equilibrated unit cell before performing production runs with MD.

Response: We thank the reviewer for the important comment. The lattice constants used in the original manuscript ($a = 20.201$ Å, $b = 19.991$ Å, $c = 13.469$ Å) was in fact taken from the literature [Vezzalini et al., Zeolites, 1997, 19, 323] studying the lattice parameters of ZSM-5 with substituted Al atoms, which is similar to the system we are studying here, but not from the IZA zeolite database without substitution as shown in the original manuscript. We apologize for this mistake and have corrected it in the revised manuscript.

Following the reviewer’s comments, we have further performed a 50-ps NPT simulation (670 K and 1 bar) to evaluate the optimized lattice constants of the HZSM-5 framework used in the manuscript. The trajectories of the lattice a , b and c were shown in Figure R2. All of the lattice parameters were found in equilibrium after 5 ps. By averaging over the range from 5 to 50 ps, the optimized lattice constants were evaluated to be $a = 20.224$ Å, $b = 20.014$ Å, $c = 13.485$ Å, which is very close to the values used in the original manuscript with a negligible discrepancy of ~ 0.02 Å (~ 0.1 %). The discrepancy is small enough and will not introduce an error through e.g. lattice strain. We have added these results in the revised manuscript to clarify this point (lines 101-102, page 4 and Supplementary Note 2).

Figure R2. The trajectories of the lattice parameters a , b and c in the NPT simulation.

- Details like the applied Gaussian basissets (DZVP, TZVP ...) for the GPW approach or the hills spawning time for the metadynamics simulations are lacking.

Response: Thank this reviewer for the helpful comment. In this work, the Goedecker-Teter-Hutter (GTH) norm-conserved pseudopotentials were used for the description of the core electrons, and the shorter range molecularly optimized GTH basis sets (denoted as dzvp-molopt-sp-gth) [VandeVondele and Hutter, J. Chem. Phys. 2007, 127, 114105] with an energy cutoff of 280 Ry was chosen to expand electronic wavefunctions. In the metadynamics simulations, the width of Gaussian hills was 0.03 and the hills spawning time step was 25 fs (50 steps). These information has been included in the revised manuscript (lines 337-340, 370-371, pages 15-16).

- Were the production runs stable enough with just a single Nosé-Hoover thermostat? As can be seen in literature, it is common to use a chain of NH thermostats.

Response: We are very grateful that the reviewer points this out. In the simulations, a Nosé-Hoover chain thermostat was used, and the length of the chain was 3, which is the default value in CP2K code. We have replaced the sentence “Nosé-Hoover thermostat was used to maintain the temperature at 670 K.” with “Nosé-Hoover thermostat with three chains was used to maintain the temperature at 670 K.” in the Method section of the revised manuscript (lines 342-343, page 15).

- How were free energies calculated from metadynamics runs?

Response: Thank the reviewer for this comment. In the simulations, we first performed several 2D metadynamics simulations, and then summed Gaussian hills to obtain the corresponding 2D free energy surfaces. Then by taking the difference between CVs, i.e. CV1 – CV2, as the reaction coordinate, we projected the 2D free energy surface onto 1D (the sampling density on the projection axis was 0.01) to further obtain the 1D free energy profiles. The methylation barrier, isomerization barrier and reaction free energy were obtained from the 1D free energy profiles. The projecting method was introduced in the work of Moors et al. [ACS Catal., 2013, 3, 2556], and the following equation was used:

$$G(\text{CV1}-\text{CV2}) = -\frac{1}{\beta} \ln \left\{ \int_{-\infty}^{\infty} d_{\text{CV1}} \exp[-\beta G(\text{CV1}-\text{CV2}, \text{CV1})] \right\}$$

We have added these information in the revised manuscript (lines 375-381, pages 16-17).

- How did the authors choose the collective variables for the free energy sampling? It also seems that throughout the studies various types of collective variables have been mixed (coordination numbers and distances). Did the authors check whether these are the appropriate CVs?

Response: We thank the reviewer for this important comment.

Regarding the MTD simulations, we used the method reported by Moors et al. [ACS Catal., 2013, 3, 2556], in which they applied the CN of C_{me}-O_{me} and C_{me}-C_b as

CV1 and CV2, respectively, to describe the benzene methylation reaction. However, the 2D free energy surface of toluene methylation reaction is much more complex than that of benzene methylation. Here the free energy surface contains four metastable states (reactant and proton-PX/MX/OX) and several transition states. It is difficult to perfectly show all states on the free energy surface through the parameter setting of 2 CV axes. After a series of tests, we selected the CN of C_{me}-O_{me} and C_{me}-C_p as CV1 and CV2, and set n_n, n_d and r₀ to 6, 12, 2.0 Å, respectively. This ensures that the states of IS, TS(R₁), TS(R₃) and proton-PX can occupy a relatively large area on the 2D free energy surface, and the reliability of their structure and free energy can be guaranteed, and confirmed by structural analysis and error analysis, as shown in the revised manuscript (see Figure 3, Supplementary Note 3 and 6).

As for the simulations of proton-xylene deprotonation reaction, this is a relatively simple reaction which only involves the migration of proton. Therefore, we used a more intuitive reaction coordinate, i.e. O_z-H_{p/m} bond length, to describe the reaction. We adopted a relatively small change rate of the reaction coordinate to ensure the reliability of test results. From the subsequent reaction path analysis, structural analysis and energy analysis results (see Figure 4 and Supplementary Note 8), we found that this reaction coordinate is appropriate and reliable.

For the diffusion simulations, Cnudde et al. [J. Am. Chem. Soc., 2020, 142, 6007] used similar reaction coordinates to study the diffusion of ethylene and propylene in SAPO-34 from one cage to an adjacent cage, and found that this method is appropriate for the modeling of diffusion within zeolites.

• I am not convinced that all reported free energies are fully converged. Therefore it is needed to report error bars. Figure S3 for example does not convincingly show that the numbers are converged (very sudden jumps etc.). What is the free energy spike in the intermediate structure in figure S1?

Response: We sincerely thank the reviewer for the comment to help us improve the manuscript. The uncertainties of free energies obtained were analyzed and the error bars were added in the figures of the revised manuscript, following the suggestions of the reviewer.

Regarding Figure S3, to verify that the free energy has converged, we continued to run the MTD-PX/MX simulations for 50 ps. Figure R3 below shows the free energy barrier and reaction free energy curves against time. According to this figure, both the barrier and reaction energy fluctuate within a reasonable range, instead of continuously rising or falling, which means that they have indeed converged.

The two metastable states in the middle of Figure S1 represent proton-MX and proton-OX in the MTD-PX simulation, respectively. Figure R4 shows the free energy change curve of proton-MX/OX against time in MTD-PX. One can find that the free energy of proton-MX/OX remains the same or suddenly decreases during simulation, suggesting that these regions are very easy to overfill and the corresponding free energy are not reliable. This figure has been included in the revised SI to clarify this point (Supplementary Note 3 and 4).

Figure R3. The free energy change curves of methylation barrier, isomerization barrier and reaction energy over time in (a) MTD-PX and (b) MTD-MX simulations.

Figure R4. The free energy change curve of proton-MX/OX against time in the MTD-PX simulation.

• Why were various types of free energy sampling methods used? Using all methods for all reactions could also give a feeling of how reliable the results are. Now the choice is not motivated and seems to be rather random. Maybe the authors had some practical reasons for this choice?

Response: Thank the reviewer for this comment. Here the free energy sampling methods were selected based on the complexity of different reaction steps. In general, the metadynamics could explore high-dimensional free energy surfaces in a convenient way, while the slow-growth and blue-moon methods used for determining one-dimensional free energy surfaces are more efficient with lower computational costs and lower fluctuation compared with standard (i.e. non-well-tempered) metadynamics simulations. For example, the methylation reaction studied in our work contains four metastable states and several transition states. For these reactions, at least two CVs should be used and therefore the metadynamics was adopted. For other reactions that are relatively simple, the slow-growth or blue-moon method were used. We have stated in the revised manuscript the reason for selecting the corresponding free energy sampling method (lines 348-353, page 16).

- The free energy curves in Figure 4 do not show a clear free energy minimum for proton-PX and proton-MX. Was the sampling sufficient? Maybe sampling at larger distances was needed as well?

Response: We thank the reviewer for the suggestion. The distance of 3.0 Å for O_z-H_p was used because 3.0 Å is far enough where there are only weak vdW interactions between H_p and acid site. To confirm 3.0 Å can represent the free energy minimum, we further performed two additional tests at larger distances for deprotonation reactions of proton-PX and proton-MX. Five parallel simulations were performed for each reaction, and the averaged free energy profile was obtained through the Jarzynski average formula. The results are shown in Figure R5 which were obtained with a reaction coordinate moving rate of $0.000109 \text{ \AA frame}^{-1}$. As for proton-PX, the free energy change is small and fluctuates around 0, and for proton-MX the free energy increases gradually, demonstrating that 3.0 Å approaches to the free energy minimum and the slow-growth simulations starting from 3.0 Å are sufficient. We have revised the manuscript to take the helpful suggestion from this reviewer into account (lines 230-234, page 11 and Supplementary Note 8).

Figure R5. Free energy profiles of (a) proton-PX and (b) proton-MX deprotonation reaction in the $O_z-H_{p/m}$ distance range of 3.0 to 4.0 Å.

- The authors state “The effective barrier of the whole reaction of toluene to p-xylene is 82.2 kJ/mol, which is close to the values of 60.52 to 86.41 kJ/mol obtained in pervious experiments and simulations”. If the authors call a difference of roughly 22 kJ/mol close, what is then considered as accurate?

Response: We thank this reviewer for the helpful comment. The values of 60.52 to 86.41 kJ/mol were obtained from the literature including both experiments and simulations. The reason we mentioned these values here was to prove that the effective barrier calculated in our work lies within the range of activation energies reported in the literature. We have changed the statement in the manuscript to avoid confusion (lines 253-254, page 12).

- Based on Figure 5, the authors conclude that the probability to form p-xylene and m-xylene is nearly identical. From a kinetic perspective this is true, however there is a

difference in reaction free energy, so from a thermodynamic viewpoint one would expect a difference (preference for p-xylene).

Response: We fully agree with the reviewer regarding this point. We have replaced the sentence “*Since the effective barriers for the formation of p-xylene and m-xylene are almost identical, the probability to form p-xylene and m-xylene in the primary product is similar.*” with “*The effective barriers for the formation of p-xylene and m-xylene are almost identical, and p-xylene is slightly more stable than m-xylene in the catalyst pores.*” in the revised manuscript (lines 266-268, page 12).

Reviewer #2 (Remarks to the Author):

The manuscript of Chen, Liu and Yang investigates the key steps in selectivity for toluene methylation, and concludes that m-xylene will preferentially isomerise to p-xylene, and then diffuse out of the catalyst zeolite, because the m-xylene cannot diffuse as easily. The claims are novel and would be interesting to a wide audience, because they offer atomistic insight into the mechanism in an industrially valuable process.

The work is convincing, though this reviewer would like to see several questions addressed for the scientific conclusions to be rigorous:

- with respect to the previous literature, the authors change their notation for the catalyst from ZSM-5 to H-ZSM-5 after paragraph 2. Is this deliberate? Have other counterions been considered previously? It would be good to clarify this point.

Response: We are very grateful that the reviewer points this out. We have checked the references and made it clear that these should be HZSM-5. We also checked through the manuscript to avoid confusion this may cause.

- The authors might want to explain why they chose to run simulations at 670 K, given experiment demonstrates the direct mechanism occurs at very slightly higher (> 673 K; is this industrial conditions?). Why not at higher temperatures? Presumably the stepwise mechanism is competitive at the chosen temperature - it would have been informative to compare both.

Response: We sincerely thank the reviewer for the important comment to help us make a clearer description. The temperature of 670 K was used because it matches the typical toluene methylation reaction temperatures that ranges from 350 to 500°C [Rabiu et al., *Ind. Eng. Chem. Res.*, 2008, 47, 39; Wang et al., *Nat. Commun.*, 2019, 10:4348; Mantha et al., *Ind. Eng. Chem. Res.*, 1991, 30, 281].

Following the very helpful suggestion regarding the possible preference of the stepwise mechanism, we further performed a series of MTD simulations to explore the free energy profiles of this mechanism. $C_{me-O_{me}}$ and C_{me-O_z} were selected as CV1 and CV2, respectively, to obtain the free energy surface of methyl groups formation at the acid site (the first step of the stepwise mechanism). Figure R6 shows the curve of obtained free energy barrier and reaction free energy against time. We can see that the free energy barrier is greater than 150 kJ mol^{-1} (although the fluctuation is relatively large), which is close to the value of 152 kJ mol^{-1} reported by Moors *et al.* [*ACS Catal.*, 2013, 3, 2556]. One can see that the free energy barrier to form the surface methyl group is much higher than the effective barrier in the concerted mechanism, and therefore the concerted mechanism should be preferred under the conditions considered in our work. We have added those results and discussions in the revised manuscript (lines 95-98, page 4 and Supplementary Note 1).

Figure R6. The curve of free energy barrier and reaction free energy of surface methyl groups formation reaction against time.

- What temperature were the crystal structure parameters obtained for? And more importantly, what cell parameters do the simulation settings return with e.g. NPT? How much discrepancy is there between the used and equilibrium lattice constants, and does this introduce an error through e.g. lattice strain?

Response: We thank the reviewer for the important comment. The lattice constants used in the original manuscript ($a = 20.201 \text{ \AA}$, $b = 19.991 \text{ \AA}$, $c = 13.469 \text{ \AA}$) was in fact taken from the literature [Vezzalini et al., Zeolites, 1997, 19, 323] studying the lattice parameters of ZSM-5 with substituted Al atoms, which is similar to the system we are studying here, but not from the IZA zeolite database without substitution as shown in the original manuscript. We apologize for this mistake and we have corrected it in the revised manuscript.

Following the reviewer's comments, we have further performed a 50-ps NPT simulation (670 K and 1 bar) to evaluate the optimized lattice constants of the HZSM-5 framework used in the manuscript. The trajectories of the lattice a , b and c were shown in Figure R7. All of the lattice parameters were found in equilibrium after 5 ps. By averaging over the range from 5 to 50 ps, the optimized lattice constants were evaluated to be $a = 20.224 \text{ \AA}$, $b = 20.014 \text{ \AA}$, $c = 13.485 \text{ \AA}$, which is very close to the values used in the original manuscript with a negligible discrepancy of $\sim 0.02 \text{ \AA}$ ($\sim 0.1 \%$). The discrepancy is small enough and will not introduce an error through e.g. lattice strain. We have added these results in the revised manuscript to clarify this point (lines 101-102, page 4 and Supplementary Note 2).

Figure R7. The trajectories of the lattice parameters a , b and c in the NPT simulation.

- Why was T12 chosen for the aluminium site? How does this influence the results? And why was this Si/Al ratio chosen?

Response: We thank the reviewer for the important comment. The T12 site is located at the intersection of the straight and sinusoidal channels and can provide maximal reaction space for the guest molecules, therefore it was widely considered in previous studies [e.g., Moors, et al., ACS Catal., 2013, 3, 2556; De Wispelaere, et al. Catal. Today, 2018, 312, 35, etc.]. In addition, toluene is a relatively large molecule (7.0 Å in diameter) compared with the channels of ZSM-5 (6.9~7.2 Å in diameter) [Kaeding et al., J. Catal., 1981, 67, 159], and it is reasonable to expect that T12 is the most suitable site for the toluene methylation reaction. It would be absolutely a good topic to be discussed in the future work investigating the role of different sites in the reactions, taking the effects of transition-state-shape selectivity into consideration. Nevertheless, in the current work, we are only considering the T12 site because this site would guarantee that there is enough space for the reaction to happen, thus we can put more focus on the intrinsic reaction kinetics of toluene methylation at easily accessible Brønsted acid sites. The manuscript has been modified to clarify this issue (lines 103-105, pages 4-5).

In this work, the orthorhombic ZSM-5 unit cell with 96 Si atoms was used. Only one Al atoms were embedded in the framework, resulting in a Si/Al ratio of 95, which is larger than the experimental values of 45 [Svelle, et al., J. Catal., 2004, 224, 115; Svelle, et al., J. Catal., 2005, 234, 385]. The value in experiments corresponds to about two Al atoms per unit cell. However, in the 96T model, when the substitution sites are far apart, it can be considered that the interaction between the two sites is very weak, thus our model would represent the active sites in a proper way.

- For Figure 2, some clarity would be good as to why formation rates of protonated xylenes would be equivalent; reviewing the diagram, the pathway seems to go through proton-OX and proton-MX, which would therefore have greater rates? This also appears the case from the diagonal CV2-CV1 plot (L138), where the OX and MX intermediates are lower in energy?

Response: We thank this reviewer for the comments. The 2D free energy surfaces (Figure 2 and Figure S2 in the original manuscript) were plotted to explore the landscape of the states of IS, proton-PX/MX/OX and related TS; while the 1D free energy profiles (Figure S1 and S2 in the original manuscript) were used to estimate the free energy values. The sequence, IS to proton-MX/OX to proton-PX, in the 1D free energy profiles (Figure S1 and S2 in the origin manuscript) does not mean the formation of proton-PX must go through proton-MX/OX. From the 2D free energy surfaces, we can see that the reaction path bifurcates from the initial transition state (TS1 in Figure R8(a) or TS(R1) in Figure R8(b)), and a selection between possible products occurs after the initial transition state. Therefore, the formation of protonated xylenes shares the same transition state; and the free energy barriers for the formation of protonated xylenes would be equivalent. We have modified Figure 2 in the revised manuscript to make a clearer description.

Editorial Note: Figure R8a is reprinted with permission from Çelebi-Ölçüm, et al., Effect of Lewis Acid Catalysts on Diels–Alder and Hetero-Diels–Alder Cycloadditions Sharing a Common Transition State, *J. Org. Chem.* 2008, 73, 19, 7472–7480, <https://doi.org/10.1021/jo801076t>. Copyright (2008) American Chemical Society

Figure R8. Bifurcating potential energy surface. (a) A generalized free energy surface. Source: (Reprinted from [Çelebi-Ölçüm, et al., *J. Org. Chem.* 2008, 73, 7472]) (b) Our results.

In addition, Figure R9 below shows the free energy change curve of proton-MX/OX against time in MTD-PX. One can find that the free energy of proton-MX/OX remains the same or suddenly decreases during simulation, suggesting that these regions are very easy to overfill and the corresponding free energy are not reliable. This figure has been included in the revised SI to clarify this point (Supplementary Note 4).

Figure R9. The free energy change curve of proton-MX/OX against time in the MTD-PX simulation.

- On L195, the TS structures are noted as being similar but C_o differs significantly. Why is this? Also close by, it might be nice to comment on the intermediate configurations in the pore (obvious from Fig S4)

Response: We sincerely thank the reviewer for the important comment to help us make a clearer description. The difference between $C_{me}-C_o$ and $C_{me}-C_p$ or $C_{me}-C_m$ is that $C_{me}-C_o$ shows a much larger fluctuation than the latter two distances, which can be attributed to the fact that the selected configurations for analysis were based on the ranges of CV $C_{me}-C_p$ or $C_{me}-C_m$. It should be mentioned here that the average distances of $C_{me}-C_o$ in

TS(R1) and TS(R2) is 2.91 and 3.07 Å, respectively, suggesting that these two structures are similar regarding this distance.

Figure S4(a) (Figure S8(a) in the revised SI) shows the structural analysis results of proton-PX basin of MTD-PX simulation, which suggests the formation of C_{me} - C_p bond in proton-PX as expected. Similarly, Figure S4(b) (Figure S8(b) in the revised SI) showing the MTD-MX simulation result confirms bond formation between C_{me} and C_m . Through this, we confirmed the structures of FS in the two MTD simulations, and provided support for the calculation of free energy differences, e.g. reaction energy. These descriptions have been added to the revised manuscript, according to the helpful suggestion from this reviewer. (Supplementary Note 6)

- On L215, discussion is given of proton migration. Can the authors confirm this is barrierless? As it isn't clear from the text and diagrams.

Response: We sincerely thank the reviewer for the important comment. After analyzing the distances between H and carbon in aromatic ring, we found that the statement made in the original manuscript is less accurate. We calculated the variation of C_m - H_p distance with time in the whole process of proton-PX dehydrogenation reaction, the results of which are shown in Figure R10. We found that, in most cases, the C_m - H_p distance is higher than the C-H bond length (1.1 Å). Therefore, the proton was actually not *transferred* to the adjacent benzylic C atom during the process. We have removed this description in the revised manuscript.

Figure R10. The trajectories of C_m - H_p distance in 10 slow-growth simulations for the process of proton-PX dehydrogenation reaction. The C-H equilibrium distance of 1.1 Å were also indicated in the figures for clarity.

- For the diffusion studies, convenience is not an appropriate justification for use of a silicate model. In reality, the aluminium are present and so here the diffusion is artificially elevated; what would be the consequence of having aluminium sites on the diffusivity, and does this alter the conclusions?

Response: We sincerely thank the reviewer for the important comment to help us improve the manuscript. We have performed new simulations with aluminum sites included to verify the consequence of the diffusivity. The aluminum atom was placed at the T12 site and the diffusion path is across the aluminum site. All other settings for

the free energies are the same as in the original manuscript. We also calculated the uncertainties of free energy barriers obtained. Firstly, the standard deviation (σ_f) of constrained force was estimated using the block average method. Then, the linear error propagation theory was used to calculate the uncertainty of free energy profiles by summing up the variance of work in each bin (σ_w^2) from minimum to maximum of the free energy profiles.

The results were shown in Figure R11 and Table R1 with uncertainty. One can find that there is no remarkable deviation compared with the results obtained from the zeolite without aluminum as reported in the original manuscript (also shown in Table R1 for comparison). The new results were added in the revised manuscript to replace the simulations obtained in zeolite without aluminum (Figure 6 and Supplementary Note 9).

Figure R11. (a-b) Possible paths for p-xylene diffusion at the intersection tunnels. Free energy profile of p-xylene and m-xylene diffusing (c) along straight channel; (d) along zigzag channel. The shaded region of each line illustrates the uncertainties.

Table R1. Diffusion barriers (in kJ/mol) of m-xylene and p-xylene along the two channels in zeolite with and without Al sites.

	Straight		Zigzag	
	with Al	without Al	with Al	without Al
m-xylene	23.2 ± 4.2	21	46.8 ± 4.4	48.2
p-xylene	19.4 ± 2.7	13.6	15.9 ± 4.6	17.5

- On L265, diffusion rates are eluded to. This should allow derivation of hopping rates that can be compared to the authors results, assuming there is a hopping mechanism?

Response: A method proposed by Maxwell-Stefan can be employed to describe the diffusion in zeolite with the distribution of sites and the hopping rates of the molecule crossing the sites. Mechanistically, the diffusivity D is related to the displacement of the nearest neighbor sites L , and the hopping rate ν , in the form of

$$D = (1/z) L^2 \nu, \quad (\text{R1})$$

where z represents the number of nearest neighbor sites and $z = 2$ in our cases. [Keil, et al., Rev. Chem. Eng. 2000, 16, 71]. With the free energy barrier G , the hopping rate was estimated by the transition state theory as

$$\nu = k_B T / h \exp(-G / k_B T), \quad (\text{R2})$$

where h is the Planck constant [J. K. Nørskov, Fundamental concepts in heterogeneous catalysis, John Wiley & Sons, Inc., New Jersey 2014]. One can find that the hopping rates are mainly related with the free energy barriers and lower barriers would give rise to higher hopping rates.

There are also aspects of the methodology and statistical analysis that the reviewer thinks could be clarified:

L64: Please define T for 4T. What is the difference between this and T12 also needs explaining briefly.

Response: We sincerely thank the reviewer for the important comment to help us make a clearer description. In the literature, to facilitate the calculations, cluster models were cut from the periodic structure of zeolites and saturated with hydrogen atoms to avoid dangling bonds. For example, the cluster models denoted as 3T and 4T suggest that there are three or four Si (Al, P, etc.) atoms in the cluster, respectively. The structure of 4T cluster model used in Ref. 16 [Arstad, et al. J. Phys. Chem. B 2002, 106, 12722] of the original manuscript is shown as Figure R12(a). The notation of T1 to T12 was used under different circumstances to differentiate the symmetrically equivalent sites in ZSM-5, which is defined in the IZA-SC Database of Zeolite Structures. The positions of T12 site of ZSM-5 are illustrated in Figure R12(b) for clarify. We have revised the manuscript to make a clearer description (lines 64-65, page 3).

Figure R12. (a) The 4T cluster used as zeolite model. Source: (Reprinted from Ref. 16 of the origin manuscript). (b) The position of T12 site of ZSM-5. Color code: blue, Si on T12; yellow, Si not at T12; red, O.

Editorial Note: Figure R12a is reprinted with permission from Arstad, B., Kolboe, S. & Swang, O. A theoretical investigation on the methylation of methylbenzenes on zeolites. *J. Phys. Chem. B* 106, 12722-12726, doi:10.1021/jp020851o (2002). Copyright (2002) American Chemical Society.

L171: "Series of configurations" - how many, and selected by what justification? Are there error bars from the sampling?

Response: We sincerely thank the reviewer for the important comment to help us make a clearer description. Following this comment, we further performed a rigorous structural analysis in the revised manuscript. The approximate areas of the IS, TS(R₁), TS(R₃) and FS (proton-PX) states were obtained in the 2-D free energy surface using a certain energy range. The energy range for the IS and FS region is from $A_{IS/FS}$ to $A_{IS/FS} + k_B T$, while for TS this is from $A_{TS} - 0.5 k_B T$ to $A_{TS} + 0.5 k_B T$ ($1 k_B T = 5$ kJ/mol at 670 K). Then rectangle regions that meet the criteria were selected, as shown in Table R2. All the frames that within rectangle regions were analyzed, and the structural analysis was performed accordingly. It should be noted that in the MTD simulation runs with historically bias potential, the number of structures can only be used to demonstrate that the selected structures were adequate, but the exact numbers obtained do not represent the equilibrium properties. These results have been added to the revised manuscript (Supplementary Note 6).

Table R2. The rectangle regions that meet the energy criteria and the corresponding number of structures in the MTD-PX simulation.

region	CV1	CV2	number of structures
IS	max: 0.880 min: 0.850	max: 0.049 min: 0.040	329
TS(R ₁)	max: 0.480 min: 0.370	max: 0.065 min: 0.039	1759
TS(R ₃)	max: 0.080 min: 0.056	max: 0.454 min: 0.408	692
FS	max: 0.078 min: 0.054	max: 0.735 min: 0.699	428

L193: "C_m is much higher than ..." - can this be made quantitative with e.g. centre of mass differences?

Response: We sincerely thank the reviewer for the important comment to help us make a clearer description. According to the above method, the structures of TS regions of R₁ and R₂ in both MTD-PX and MTD-MX simulations can be obtained. Table R3 shows the average distances of $C_{me}-C_p/C_m/C_o$ and $C_{me}-O_{me}$ of TS(R₁) and TS(R₂) structures in both MTD-PX and MTD-MX simulations. These results suggest the TS(R₁) and TS(R₂) structures in the two MTD simulations are very similar, i.e., C_{me} is closer to C_m and C_o than to C_p , and the $C_{me}-O_{me}$ distances are almost identical.

Table R3. Average distances of $C_{me}-C_p/C_m/C_o$ and $C_{me}-O_{me}$ of TS(R₁) and TS(R₂) structures in both MTD-PX and MTD-MX simulations, with uncertainty.

TS (R ₁ /R ₂)	C _{me} -C _p (Å)	C _{me} -C _m (Å)	C _{me} -C _o (Å)	C _{me} -O _{me} (Å)
MTD-PX	3.28±0.09	2.83±0.33	2.91±0.59	2.10±0.05
MTD-MX	3.60±0.47	3.22±0.09	3.07±0.52	2.11±0.02

Fig6: Are there error bars for the free energy landscapes?

Response: We thank for the reviewer's suggestion. The uncertainties of free energy landscapes were analyzed and the error bars were added in Supplementary Note 3, following the suggestions of the reviewer.

L315: Why did the authors chose the exchange-correlation functional, dispersion corrections, and timestep chosen for the MD?

Response: We hereby summarize the reasons for settings of the computational approach as follows:

Exchange-correlation functional: The revPBE functional was selected due to its improved ability to describe energies related to catalytic processes compared with the commonly used PBE functional for solid-state calculations [Yang, et al., J. Chem. Phys., 2010, 132, 164117]. This functional was widely used in recent theoretical studies of zeolite-catalyzed reactions [e.g. Moors, et al., ACS Catal., 2013, 3, 2556; De Wispelaere, et al., Catal. Sci. Technol., 2016, 6, 2686.; De Wispelaere, et al., ACS Catal., 2016, 6, 1991; Martínez-Espín, et al., ACS Catal., 2017, 7, 5773, etc.].

Dispersion corrections: We note that DFT often provides a poor description of van der Waals dispersive interactions [Grimme et al., J. Chem. Phys., 2010, 132, 154104]. The long-range dispersive interactions that can impact predictions, especially for weakly interacting adsorbates with metal surfaces. A number of correction schemes have been proposed to improve the description of dispersive interactions. In particular, the D3 method by Grimme has been found to be useful. The Grimme's DFT-D3 empirical correction was widely used in recent theoretical studies of zeolite-catalyzed reactions to illustrate the long-range dispersion interactions between the adsorbates and catalysts [e.g. Van Speybroeck, et al., J. Am. Chem. Soc., 2011, 13, 888; Moors, et al., ACS Catal., 2013, 3, 2556; De Wispelaere, et al. Catal. Today, 2018, 312, 35; Lambert, et al., J. Phys. Chem. C, 2019, 124, 11469, etc.].

Timestep: The fast vibration in the system is X-H stretching (X = O, C, etc.) with a period of about 10 fs (i.e., ~3000 cm⁻¹ in infra-red spectrum experiments). Therefore, a timestep of 0.5 fs was chosen on the basis of the X-H stretching period. A larger timestep would produce significant errors or make the simulations unstable.

We have revised the manuscript to clarify these points (lines 335-337, page 15).

L334: How was the value of r₀ chosen?

Response: Thank this reviewer for the comment. In this study, the proper value of r₀ was selected to make sure that the position of each intermediate state and transition state can be identified clearly from the 2D free energy surfaces. We have made a comparison of using r₀ value of 1.59 Å, which was used in ACS Catal., 2013, 3, 2556, for the study of benzene methylation, and 2.0 Å in Figure R13 below. When 1.59 Å was used as r₀, we can see that several stable-state regions are very small and difficult

to distinguish from each other. When 2.0 Å was used as r_0 , as shown in Figure R13(b), we can clearly distinguish the position of each stable-state and transition state. Moreover, structural analysis, free energy analysis and error analysis all proved the credibility of the MTD simulation results.

Figure R13. The 2D-FES of MTD-PX simulations when (a) 1.59 Å or (b) 2.0 Å is chosen as r_0 .

L356: Naive question, but Eq2 doesn't show obviously how the averaging is achieved - no summation?

Response: Eq. 2 (the Jarzynski equation) is written as $\Delta A = -\beta^{-1} \ln \overline{\exp(-\beta W)}$ in our manuscript, according to Jarzynski's original paper [Jarzynski, Phys. Rev. Lett. 1997, 78, 2690]. In this equation, the expression of \bar{x} denotes a canonical averaging over x , i.e., $\bar{x} = \frac{\sum_{i=1}^n x_i}{n}$.

L369: "l-value" doesn't have a definition of "l"?

Response: We thank this reviewer for pointing this out. In the manuscript, l is the distance between the center of xylenes and the center of the zeolite and was defined as the CV. We have explicitly defined " l " in the revised manuscript (line 402, page 17).

L375: Why the last 4 ps of the trajectory?

Response: Thanks for the comment. This is because the system needs some time to reach thermal equilibrium from the initial input structure, some trajectories at the beginning of the constrained MD simulations should be omitted when collecting the data. Two examples of trajectories F_c were given in Figure R14. From the trajectories of constraint force F_c , we can see that F_c reached equilibrium after the first 1000 steps, suggesting that considering the following 4000 steps is statistically enough. Therefore, in each constrained MD simulation, totally 5000 steps were performed and the last 4000 steps of the trajectory were analyzed. We note that there is a mistake on the conversion from step to time in the original manuscript. Totally 5000 steps were performed in each simulation with a timestep of 0.5 fs, thus the simulation time is 2.5 ps. This mistake would not impact the main results of this work, and we have corrected this in the revised manuscript. (lines 410-412, page 18 and Supplementary Note 9)

Figure R14. The trajectories of constraint force F_c in the constrained MD simulations. Two trajectories, i.e., p-xylene diffusion along straight channel at $l = 1.0 \text{ \AA}$ (a) and 2.0 \AA (b), were given as examples. The shaded regions denote that the first 1000 steps were omitted.

Finally, minor spelling/grammatical mistakes, clarifications/corrections/improvements to images, and inconsistencies in the bibliography presentation are included in the attached PDF as comments.

Response: We would like to thank the reviewer for helping us avoid these mistakes/typos. The manuscript has been checked through to take these suggestions into account.

Reviewer #3 (Remarks to the Author):

The authors present a computational study into the Methylation of toluene with methanol to produce p-xylene. The authors use ab initio molecular dynamics (AIMD) simulations and well-established free energy sampling methods.

My expertise is in the development and application of enhanced sampling methods so my comments will be mostly focusing on this part of the manuscript.

Overall, the methodology used in the paper is appropriate for tackling the scientific question that the authors want to answer. The simulations seem to be performed properly though some of analysis could be done better, see my comments below.

I believe that the manuscript can eventually be accepted for publication in Nature Communication. However, at the moment, I find various issues with manuscript that need to be addressed before it can be accepted for publication.

I would need to see the manuscript again, but once the authors have addressed the issues raised in my report, I can very likely recommend that that manuscript is accepted.

Please find in the following my comments:

* (Page 1, line 13-14) "and state-of-the-art free energy sampling methods" I would not call the free energy methods used in the paper to be state of the art. Instead they are all well established and proven methods for free energy calculations. Thus, I would ask the authors to change this statement.

Response: We sincerely thank the reviewer for the suggestion to help us make a clearer description. In the original manuscript, the statement of "state-of-the-art" would be referred to the technology of DFT-based AIMD simulations associated with free energy sampling methods. According to the reviewer's suggestion, we have removed the statement of "state-of-the-art" in the revised manuscript.

* Page 4, line 91 and page 5, line 116) The acronym PX, MX, and OX are never explicitly defined. This should be clarified.

Response: We sincerely thank the reviewer for this suggestion. The acronym PX, MX, and OX were defined in the first paragraph, and the acronym proton-PX and proton-MX were also defined when they first appeared. The usage of these acronyms in other parts of the revised manuscript were also checked and corrected for consistency.

* (Page 7) The authors perform two metadynamics simulations, MTD-PX and MTD-MX. Apart from the statement, "In order to distinguish the proton-MX basin from the overlapping area, we further performed MTD simulations on the reaction of proton-MX (MTD-MX) generation from methanol and toluene.", I find it not well explained the reason for doing the second MTD-MX simulation. I would understand this is due to the choice of the CVs. Since many reader will not be experts on enhanced sampling simulations, they might not understand this issue. Therefore, this should be discussed more clearly. Furthermore, it should be briefly explained in this section how these two simulations differ and what makes the MTD-MX simulation more suited to describe the reaction of proton-MX.

Response: We are very grateful that the reviewer points this out. At the very beginning of this study, we tried to use one MTD simulation (i.e., MTD-PX) to obtain the 2D free energy surface and to explore the reactivities of all the related reactions. However, four metastable states (reactant, proton-PX/MX/OX) and several transition states appeared on the 2D free energy surface, and it was difficult to separate all these states. The regions of proton-MX and proton-OX are close to each other and not well-separated, especially when projecting the 2D free energy surface into 1D. Figure R15 below shows the free energy change of proton-MX and proton-OX against simulation time. One can find that the free energy of proton-MX/OX remains the same or suddenly decreases during simulation, suggesting that these regions are very easy to overflow and the corresponding free energy are not reliable. Therefore, to ensure that the states of IS, TS(R₁), TS(R₃) and proton-MX have relatively large area on the 2D free energy and are easily separated from other states, we chose the CN of C_{me}-O_{me} and C_{me}-C_m as CV1 and CV2, respectively, and performed another MTD simulation (namely MTD-MX). From the MTD-MX simulation, we can obtain the accurate free energy surface for the production of proton-MX. We have revised the manuscript to clarify this point (lines 155-157, page 7 and Supplementary Note 4).

Figure R15. The free energy change curve of proton-MX/OX against time in the MTD-PX simulation.

* (Page 7) One issue with standard (i.e. non-well-tempered) metadynamics simulations, like the authors employ, is that it is hard to know when to stop the simulation. This is what the authors discuss on lines 156-167. Therefore, a common solution is to employ a time-averaged estimate of the free energy surface, see Equation 9 in <http://doi.org/10.1038/s42254-020-0153-0>. In this way, one can also assign an error to the free energy estimate. This would be a more appropriate way to analyse the metadynamics simulations and I would strongly encourage the author to perform such analysis. At the very least, they should acknowledge that it is possible to perform such analysis.

Response: We sincerely thank the reviewer for the precious suggestion to help us improve the manuscript. Using the method recommended by the reviewer [in Bussi et

al., Nat. Rev. Phys., 2020, 2, 200], the uncertainties of free energy were analyzed for our metadynamics simulations. We obtained the average 1D free energy profile (named averaged 1D FEP) after t_{fill} that all the free-energy minima are filled with Gaussian hills, then computed the values and uncertainties of the free energy barrier and reaction free energy. According to the barrier and reaction energy curves shown in Figure R16, we determined the t_{fill} (60 ps for MTD-PX simulation and 80 ps for MTD-MX simulation). Block average method were used with 4 ps in each block to estimate the uncertainties of the barrier and reaction energy. All these results are shown in Figure R17 and the error interval is acceptable. The IS was used as the reference point to confirm the uncertainties of the methylation barrier and reaction energy; the FS was used as the reference point to calculate the error of the isomerization barrier (lines 147-149, 169-173, page 7 and Supplementary Note 3).

Figure R16. Methylation barrier, isomerization barrier and reaction free energy against time in (a) MTD-PX and (b) MTD-MX simulations.

Figure R17. Average 1D free energy profiles with error bars of (a) MTD-PX (IS as reference point); (b) MTD-PX (proton-PX as reference point); (c) MTD-MX (IS as reference point); (d) MTD-MX (proton-MX as reference point) simulations.

* (Page 7) Following at the previous points. For the metadynamics simulations, the authors should assign error bars to their results and values. For example, for Figure S1 and S2 (lower panel) in the SI, it would be good to get a feeling for the error bars.

Response: We sincerely thank the reviewer for the suggestion to help us improve the manuscript. According to the suggestion, the uncertainties of free energy were analyzed for the metadynamics simulations, and the error bars were assigned in the related figures.

* (Page 7 lines 156-157) This statement, "It should be noted some fluctuations on the energies calculated." This is strictly only true for standard (i.e. non-well-tempered) metadynamics simulations. The authors should modify the statement to clarify this.

Response: We thank the reviewer for the suggestion to help us make a clearer description. We have modified the statement with "*free energy in standard (i.e. non-well-tempered) MTD simulation*", according to the suggestion (lines 165, page 7).

* (Page 7, lines 165-167) I find the following statement to be unclear: "In the following discussion, the free energy differences were taken from the 1D potential energy surface obtained by the sum of all Gaussian hills." The authors employ a 2D metadynamics simulations. Thus from the sum of the Gaussian hills, they obtain a 2D free energy surface. Then they can perform a 1D projection of the 2D FES. I assume this is what they discussing here, however, I find it badly worded and this should be clarified. Furthermore, what is the collective variable used for the 1D FES? That is not clear. If it is the diagonal CV2-CV1 like mentioned on line 138, this should be mentioned again here.

Response: We thank the reviewer for the invaluable suggestion to help us make a clearer description. The reviewer is completely right. In our work, the 1D free energy profiles were obtained by projecting the 2D FES, and the collective variable used for the 1D free energy profile was CV1-CV2, not CV2-CV1 as mentioned in the original manuscript. We apologize for this typo and mistake made while preparing the original manuscript. In the revised manuscript, we calculated the free energy barrier and reaction energy from the averaged 1D free energy profiles, following the suggestion of this reviewer. We have clarified this point and make a clearer description on this statement (line 178, page 8).

* Following on the previous point. In multiple places the authors incorrectly use "potential energy surface" when they are referring to a "free energy surface", for example lines 166, 172. The authors need to go through the manuscript and check that this is corrected everywhere.

Response: We thank the reviewer for pointing out this. We have carefully checked through the manuscript to correct this mistake.

* (Figure 3, page 8) The frequency distribution for the distances shown in Figure 3 are

from metadynamics simulations and thus biased, unless the authors have performed reweighted when generating the distribution. This should be clearly stated in the caption of Figure 3. Furthermore, in the discussion of "Structural analysis of the MTD simulation results", it should be clearly stated that the analysis is done on biased simulation trajectories and thus the numbers obtained do not represent the equilibrium properties. However, it is fine to use the analysis to compare two different metadynamics simulation like the authors do. They should just state more clearly the limitations of the analysis and results.

Response: We sincerely thank the reviewer for the important suggestion to help us make a clearer description. According to the suggestion of the reviewer, we note that in MTD simulation runs with historically bias potential, the number of structures can only be used to demonstrate that the selected structures were adequate, but the numbers obtained do not represent the equilibrium properties. In addition, we accepted the helpful suggestion from this reviewer and reviewer #2, and adopted a more rigorous approach to perform the structural analysis. The approximate areas of the IS, TS(R₁), TS(R₃) and FS (proton-PX) states were obtained in the 2-D free energy surface using a certain energy interval as the criterion. The energy interval for the IS and FS region is from $A_{IS/FS}$ to $A_{IS/FS} + k_B T$, while for TS is from $A_{TS} - 0.5 k_B T$ to $A_{TS} + 0.5 k_B T$ ($1 k_B T = 5$ kJ/mol at 670 K). Then rectangle regions that meet the criteria are selected. All the frames that within rectangle regions were analyzed and, and the structural analysis was performed. As the energy interval is only $1 k_B T$, which is smaller enough and thus the structural analysis of the MTD simulation is credible even through the sampling is done on biased simulation (lines 210-213, page 9 and Supplementary Note 6).

* (Page 9-10) One issue with using Jarzynski's equality to obtain free energy profiles from slow-growth simulations (i.e, steered-MD) is that one generally needs a large amount of independent simulations to get converged results. Can the authors comments on this since they only perform 5 simulations? Why did they conclude that this is sufficient?

Response: We sincerely thank the reviewer for the comments to help us improve the manuscript. Following the comments, we further performed 5 additional slow-growth simulations. Totally 10 independent trajectories were analyzed with Jarzynski equality to estimate the free energy profiles, and the results are shown in Figure R18. We calculated the standard deviation of works in each individual trajectory as the uncertainty of free energy. The values of free energy and the standard deviation of works are also shown in Figure R18. For comparison, the free energy profiles obtained from 5 independent trajectories are presented in Figure R19. The difference between these two simulations is rather small and within the calculated uncertainty, thus the conclusions drawn in the current work will not be varied. We have included the results with 10 trajectories in the revised manuscript and shown that these simulations are sufficient to draw solid conclusions (see Supplementary Note 8).

Figure R18. Free energy profiles of proton-PX (a) and proton-MX (b) deprotonation reaction obtained from 10 independent trajectories. Thin lines are the work curves of individual trajectories and the thick line is the free energy profile obtained from Jarzynski equality. The values of free energy and the standard deviation of works are also shown in the figure.

Figure R19. Free energy profiles obtained from 5 independent trajectories. This figure is taken from the original manuscript and shown here for comparison.

* (Figure 4, page 9) Following on the previous point. In Figure 4, the authors only report the average energy values of the p-xylene and the TS. It would good if they also assign an error bar to the value, which they can obtain by looking at the variance/spread between the 5 different slow-growth simulations.

Response: We sincerely thank the reviewer for the suggestion. According to the suggestion, we calculated the standard deviation of works in each individual trajectory as the uncertainty of free energy. The error bars and the average energy values of free energy are shown in Figure R18, and have been added in the revised manuscript.

* (Figure 4, page 9) To make the manuscript clearer, it should be explicitly stated in the caption of Figure 4 that the results are obtained with slow-growth simulations.

Response: We sincerely thank this reviewer for the suggestion. Following the suggestion, we have revised the caption of Figure 4 to mention that the results were obtained with slow-growth simulations.

* (Page 12, lines 258-261) "...channel is sampled until the maximum free energy, corresponding to the free energy of transition states, was reached." In the blue moon sampling simulations, how do the authors detect the maximum free energy is reached? This is unclear to me and needs to be clarified in the manuscript.

Response: We sincerely thank the reviewer for the important suggestion to help us make a clearer description. Considering the translational symmetry of the system, only a half of the diffusion path, i.e., from a channel intersection along one channel to the center of the channel, were evaluated. Therefore, from the free energy profiles, the minimum and maximum can be read. We have revised the manuscript to clarify this point (lines 278-280, page 13).

* (Page 12) In the "Diffusion of xylene in ZSM-5 channels" section, the authors should explicitly state that the results are obtained using the blue moon sampling methods. Also in the caption of Figure 6.

Response: We sincerely thank the reviewer for the suggestion. Following the suggestion, we have revised the caption of Figure 6 and corresponding sentences in the "Diffusion of xylene in ZSM-5 channels" section to mention that the results are obtained with blue moon sampling methods.

* (Page 14) Do the authors have some motivation for using the revPBE functional? Has it been shown to well describe this system? If so, it might be good to mention it in the Methods section on page 14.

Response: We sincerely thank the reviewer for the suggestion. The revPBE functional was selected due to its improved ability to describe energies related to catalytic processes compared with the commonly used PBE functional for solid-state calculations [Yang, et al., J. Chem. Phys., 2010, 132, 164117]. This functional was widely used in recent theoretical studies of zeolite-catalyzed reactions [e.g. Moors, et al., ACS Catal., 2013, 3, 2556; De Wispelaere, et al., Catal. Sci. Technol., 2016, 6, 2686.; De Wispelaere, et al., ACS Catal., 2016, 6, 1991; Martínez-Espín, et al., ACS Catal., 2017, 7, 5773, etc.]. Following this suggestion, we added the motivation for using the revPBE functional in the Methods section of the revised manuscript (lines 335-337, page 15).

* (Page 14) Various details about the setup of the CP2K seem to be missing. For example, the exact type of basis set used (e.g., DZVP-..., etc). Also any cutoffs, if used. Given that there should be enough information to reproduce the results, I ask the authors to add all the relevant information about the setup of the CP2K simulations.

Response: Thank this reviewer for the helpful suggestions. In this work, the Goedecker-Teter-Hutter (GTH) norm-conserved pseudopotentials were used for the description of the core electrons, and the shorter range molecularly optimized GTH basis sets (denoted as dzvp-molopt-sp-gth) [VandeVondele and Hutter, J. Chem. Phys., 2007, 127, 114105] with an energy cutoff of 280 Ry was chosen to expand electronic wavefunctions. In the metadynamics simulations, the width of Gaussian hills was 0.03 and the hills spawning time step was 25 fs. We have included all these information in the revised manuscript (lines 337-340, 370-371, pages 15-16).

* (Page 14, line 318) The citation for the Nose-Hoover thermostat used in CP2K is missing. This should be added.

Response: We sincerely thank the reviewer for the suggestion, and the citations for the Nose-Hoover thermostat used in CP2K [Nosé, *Mol. Phys.*, 1984, 52, 255; Martyna, et al., *J. Chem. Phys.*, 1992, 97, 2635] are added in the revised manuscript as Refs. 41 and 42.

* (Page 15) The overall simulation time of the metadynamics simulations is missing. This needs to be added.

Response: We sincerely thank the reviewer for the suggestion to help us to improve our manuscript. The overall simulation time of MTD-PX and MTD-MX simulations were 145 ps and 165 ps, respectively. According to the suggestion, we have added this information in the revised manuscript (lines 372-374, page 16 and Supplementary Note 10).

* (Page 15) "The height of Gaussian hills used in the MTD simulations was 2.6255 kJ/mol (0.001 hartree) initially, then reduced to 1.3128 and 0.6564 kJ/mol to improve the accuracy of the results." The scheduling policy used here should be explicitly stated, in other words, for how long time was each height value used. For example, for first X ps 2.6255 used, then 1.3128, etc. I remind the authors again, there should be enough computational details included such that others can try to reproduce the results.

Response: We sincerely thank this reviewer for the suggestion to help us improve the manuscript. The trajectories of CV1-CV2 and the moments of reducing the Gaussian hill of MTD-PX and MTD-MX simulations are shown in Figure R20.

- The simulation time of MTD-PX was 145 ps in total, and in the simulation, the height of Gaussian hill reduced for the first time at 15 ps (2.6255 to 1.3128 kJ mol⁻¹, corresponding to 0.001 to 0.0005 Hartree), and for the second time at 41.5 ps (1.3128 to 0.6564 kJ mol⁻¹, corresponding to 0.0005 to 0.00025 Hartree).

- The simulation time of MTD-MX was 165 ps in total, and in the simulation, the height of Gaussian hill reduced for the first time at 40 ps (2.6255 to 1.3128 kJ mol⁻¹, corresponding to 0.001 to 0.0005 Hartree), and for the second time at 65 ps (1.3128 to 0.6564 kJ mol⁻¹, corresponding to 0.001 to 0.0005 Hartree).

According to the suggestion, we have added these computational details in the revised manuscript and SI (lines 372-374, page 16 and Supplementary Note 10).

Figure R20. The trajectories of CV1-CV2 and the moments to reduce the height of the Gaussian hill in (a) MTD-PX and (b) MTD-MX simulations.

* (Page 15, lines 343-345) "In addition, quadratic walls were used to restrict the CVs on the regions of free energy surface of interest (see Table S1)." I am rather confused by the usage of attractive and repulsive for the direction in Table S1. I assume that the authors mean upper and lower walls that are only active once the CV have reached a certain value to limit the CV space explored by the metadynamics simulations. The usage attractive and repulsive might indicate that the walls are always active. This needs to be clarified. I would recommend the authors to use instead upper/lower for the wall type.

Response: We sincerely thank the reviewer for the suggestion to help us make a clearer description. The reviewer's suggestion is completely right and we have clarified those terms according to the suggestion to use upper/lower for the wall type in the revised SI.

* (Page 15) To clarify, did the authors use the internal metadynamics code in CP2K? Or did they use an external code like PLUMED? If they did use PLUMED, this should be explicitly stated.

Response: We thank the reviewer for the comment to help us make a clearer description. The metadynamics simulations in this work were performed with the internal metadynamics code in CP2K, and no external library/code were used.

* (Page 15-16) The same goes for the slow-growth and blue moon sampling simulations.

Did the authors use just CP2K and the free energy methods implemented in the CP2K code? Or did they use any external library/code? Any external library/code apart from CP2K should be mentioned.

Response: We thank the reviewer for the comment to help us make a clearer description. The slow-growth and blue moon sampling simulations in this work were performed with the methods implemented in the CP2K code, and no external library/code were used.

This is Omar Valsson, Max Planck Institute for Polymer Research

Yours sincerely,

Bo Yang

REVIEWERS' COMMENTS

Reviewer #1 (Remarks to the Author):

The manuscript by Yang and co-workers has been revised thoroughly and many clarifications have been added. The authors undertook an enormous effort (including a set of additional simulations) to tackle all comments and questions raised during my first review. After carefully reading the response of the authors I am satisfied with the current version of the manuscript. Therefore, I recommend publication.

This is Kristof De Wispelaere (Ghent University).

Reviewer #2 (Remarks to the Author):

The reviewers have addressed all previous queries competently, and no further changes are requested. The manuscript offers original work that will be of interest to a broad community, given the industrial value of xylene, and now seems suitable for publication.

Reviewer #3 (Remarks to the Author):

Looking at the response of the author, I believe that the authors have addressed my comments. Therefore, I believe that the manuscript can be accepted for publication.

Minor comments:

* Page 8, Line 188-190: "The distance between Cme and Ome atoms and the distance between Hme and Oz atoms were summarized in Figure 3(a) and (b). The same quadratic walls were added for Hz-Ome and Hme-Ome to ensure that methanol is in a protonated state (Table S4), " The second sentence here is rather unclear. What does "The same quadratic walls " refer to?

REVIEWERS' COMMENTS

Reviewer #1 (Remarks to the Author):

The manuscript by Yang and co-workers has been revised thoroughly and many clarifications have been added. The authors undertook an enormous effort (including a set of additional simulations) to tackle all comments and questions raised during my first review. After carefully reading the response of the authors I am satisfied with the current version of the manuscript. Therefore, I recommend publication.

This is Kristof De Wispelaere (Ghent University).

Response: We sincerely thank the reviewer for this positive comment and helpful suggestions.

Reviewer #2 (Remarks to the Author):

The reviewers have addressed all previous queries competently, and no further changes are requested. The manuscript offers original work that will be of interest to a broad community, given the industrial value of xylene, and now seems suitable for publication.

Response: We are very grateful that the reviewer raised positive comments.

Reviewer #3 (Remarks to the Author):

Looking at the response of the author, I believe that the authors have addressed my comments. Therefore, I believe that the manuscript can be accepted for publication.

Minor comments:

* Page 8, Line 188-190: "The distance between C_{me} and O_{me} atoms and the distance between H_{me} and O_z atoms were summarized in Figure 3(a) and (b). The same quadratic walls were added for H_z-O_{me} and H_{me}-O_{me} to ensure that methanol is in a protonated state (Table S4), " The second sentence here is rather unclear. What does "The same quadratic walls " refer to?

Response: We would like to thank the reviewer again for helping us to improve the manuscript. Here the second sentence is showing that the quadratic walls added for H_z-O_{me} and H_{me}-O_{me} were the same to ensure that methanol is in a protonated state (Supplementary Table 4), and H_z is equivalent to H_{me} in this simulation. We have modified this sentence to make it clearer in the revised manuscript (page 9).